

# Further validation of the McClear estimates of the downwelling solar radiation at ground level in cloud-free conditions: The case of the Sub-Saharan Africa and Maldives Archipelago

William Wandji Nyamsi[1,3,4], Yves-Marie Saint-Drenan[2], Antti Arola[1], Lucien Wald[2]

[1]Finnish Meteorological Institute, 70211 Kuopio, Finland
[2]MINES PSL, Centre O.I.E., 06904 Sophia Antipolis, France
[3]Finnish Meteorological Institute, 00560 Helsinki, Finland
[4]Department of Physics, Faculty of Science, University of Yaoundé I, P.O. Box 812 Yaoundé, Cameroon

*Correspondence to*: William Wandji Nyamsi (william.wandji@fmi.fi)

**Abstract.** Being itself part of the Copernicus Atmosphere Monitoring Service (CAMS), the McClear service provides estimates of the downwelling shortwave irradiance and its direct and diffuse components received at
ground level in cloud-free conditions, with inputs on ozone, water vapor and aerosol properties from CAMS. McClear estimates have been validated over several parts of the world by various authors. This article makes a step forward by comparing McClear estimates to measurements performed at 44 ground-based stations located in the Sub-Saharan Africa and Maldives Archipelago in the Indian Ocean. The global irradiances $G$ and its direct component at normal incidence $B_N$ from McClear-v3 were compared to 1 min measurements made in cloud-free
conditions at the stations. The correlation coefficient is greater than 0.96 for $G$ whereas it is greater than 0.70 at all stations but five for $B_N$. The mean of $G$ is correctly estimated at stations located in arid climates (BSh, BWh, BSk, BWk) and temperate climates without dry season and hot or warm summer (Cfa, Cfb) or with dry and hot summer (Csa) with a relative bias in the range [−1.5, 1.5] %. It is underestimated in tropical climate of monsoon type (Am) and overestimated in tropical climate of savannah type (Aw) and temperate climates with dry winter
and hot (Cwa) or warm (Cwb) summer. McClear tends to overestimate the means of $B_N$. The standard deviation of errors for $G$ ranges between 13 W m$^{-2}$ (1.3 %) and 31 W m$^{-2}$ (3.7 %) and that for $B_N$ ranges between 31 W m$^{-2}$ (3.0 %) and 70 W m$^{-2}$ (7.9 %). Both offer small variations in time and space. A review of previous works reveals no significant difference between their results and ours. This work establishes a general overview of the performances of the McClear service.

## 1 Introduction

Solar radiation received at ground is the main driver behind the weather and climate systems on the planet. It is one of the essential variables in climate (Bojinski et al., 2014; Lean and Rind, 1998), air quality (GEO, 2010), terrestrial and marine environment (Dantas de Paula et al., 2020; GEO, 2014), or renewable energies (Ranchin et
al., 2020). Furthermore, it has impact on human health (Juzeniene et al., 2011) and many other aspects of our daily lives and our activities as shown by the many examples given in Lefèvre et al. (2014) or Wald (2021). The



density of power received from the sun on a horizontal surface at ground level and integrated over the shortwave portion of the electromagnetic spectrum (240-4606 nm) is called here the surface solar irradiance, abbreviated as SSI. Other terms may be found in the literature, such as solar exposure, solar insolation, solar flux, downwelling

solar irradiance at the surface, downwelling shortwave flux, or surface incoming shortwave radiation. The SSI is the sum of its direct and diffuse components. Briefly speaking, the radiation appearing to come from the direction of the sun is the direct component, noted $B$, and the diffuse component gathers the photons coming from the other directions of the sky, noted $D$. When it is necessary to clearly distinguish the SSI from its components, the SSI is called global SSI, often noted $G$, with $G=B+D$. Researchers in solar energy often termed

the SSI as global horizontal irradiance, where horizontal means horizontal surface, abbreviated as GHI (see e.g., Sengupta et al., 2021).

Of particular interest here is the SSI received in cloud-free conditions and often termed clear-sky SSI. It depends on the date and time of the day and geographic coordinates because these variables define the solar radiation impinging on a horizontal surface at the top of the atmosphere and the solar zenithal and azimuthal angles. It also

depends on the concentrations of gases, though it is generally sufficient in the case of total radiation to consider the ozone and water vapor contents only, which are very variable, and to prescribe the concentrations of the other gases to standard values. As absorption depends on local conditions of temperature, density, and pressure, the vertical profile of these variables, as well as that of the volume mixing ratio of absorbing gases excluding ozone and water vapor, must be known. This profile also allows the calculation of the scattering effects of air

molecules. The clear-sky SSI also depends on the optical properties of aerosols, which are highly variable in space and time. Usually, classes of aerosols are used, for example, sea salt, or soot, which have been assigned average optical properties. The aerosol load is given by an aerosol optical depth, known at one or more wavelengths, for example, 550 and 1020 nm, or else the optical depth at a given wavelength and Ångström exponent. Finally, the clear-sky SSI depends on the elevation of the ground above the mean sea level and on the

reflective properties of the ground.

In their extensive numerical study with two different atmospheric radiative transfer models, Oumbe et al. (2014) found that the SSI in cloudy conditions may be accurately approximated by the product of the clear-sky SSI by a cloud modification factor, also known as the clear-sky index, which does not depend on the properties of the cloud-free atmosphere. The error made in using this approximation is similar to the typical uncertainty

associated with the most accurate pyranometers, except in the case of ground albedo greater than 0.7 where the error is greater. This result underlines the importance of accurate calculation of the clear-sky SSI. A model estimating the clear-sky SSI is called a clear-sky model. It provides realistic upper limits of the SSI and contributes to quantify the radiative effects of the clouds.

There are many clear-sky models described in the scientific literature (see e.g. Gueymard, 2012; Sengupta et al.,

2021; Sun et al., 2019, 2021; Yang, 2020). The McClear model is one of them. It has been developed under the auspices of the European Commission to support the Copernicus Atmosphere Monitoring Service (CAMS) service delivering the solar radiation at ground in all sky conditions (Qu et al., 2017; Schroedter-Homscheidt, 2019). The original McClear model described in Lefèvre et al. (2013) was set into operation in 2012. After slight changes in 2013 (version v2), the current version v3 was introduced in 2018 (Gschwind et al., 2019). Though it

can be used as stand-alone model, McClear is mostly being used in synergy with the 3-h estimates of aerosol properties and daily total column contents of water vapor and ozone provided by CAMS as inputs. The McClear



service is the combination of McClear and CAMS (Schroedter-Homscheidt, 2019). It delivers time series of SSI and its direct and diffuse components at any site in the world and for any period from 2004 to date with 2 days delay for the summarizations of 1 min, 15 min, 1 h, 1 day, and 1 month.

The McClear service has thousands of users, including academics, researchers, consultants and companies in various domains (Gschwind et al., 2019). Its outputs are regularly confronted with ground-based measurements of SSI made by pyranometers and pyrheliometers by either the team in charge of its development or by these users who provide valuable direct and indirect validations of the McClear service and feedbacks on its limitations. Such validations provide valuable information on the uncertainties of the outputs of the McClear

service to non-expert users, and indirectly give information on the aerosol properties given by CAMS in areas not covered by stations measuring aerosols properties such as the AErosol RObotic NETwork (AERONET). Several validations have been reported in the scientific literature dealing with many stations in various climates but the cases of the Sub-Saharan Africa and the Western Indian Ocean were so far hardly addressed. Gschwind et al. (2019) and Lefèvre et al. (2013) performed comparisons at stations located over the whole world but these

regions. Sun et al. (2019, 2021) also dealt with the whole world and included measurements from the Southern African Universities Radiometric Network (SAURAN) in Southern Africa. Yang (2020) dealt with stations in North America; Ceamanos et al. (2014) or Ineichen (2016) used stations in Europe, Israel and Algeria, with Ineichen using one station at Mount Kenya (Kenya) and another one at Skukuza (South Africa). Other authors dealt with more local networks. Antonanzas-Torres et al. (2019) used two European stations while Lefèvre and

Wald (2016), Eissa et al. (2015a, 2015b), analyzed the consistency of performances of McClear between several close stations in Israel, United Arab Emirates and Egypt respectively. Cros et al. (2013) studied the case of La Réunion and Corsica Islands and French Guiana. Dev et al. (2017) and Zhong and Kleissl (2015) performed comparison in Singapore and California respectively. Chen et al. (2020) studied the case of the megacity Shanghai and Alani et al. (2019) focused on Morocco. Mabasa et al. (2021) assessed the quality of McClear

using 13 stations of the South African Weather Services in South Africa.

The purpose of this article is to expand knowledge or strengthen existing knowledge in Sub-Saharan Africa and Western Indian Ocean. More exactly, it aims at adding to the continuous documentation of the validation of the McClear service by performing a comparison between its outputs and measurements made at stations in Botswana, Kenya, Malawi, Namibia, Senegal, South Africa, Tanzania, Uganda, and Zambia. Also included are

three additional stations located in the Maldives Archipelago situated in the Indian Ocean. A secondary goal is to assess whether our findings are in agreement with similar published works regarding the range of values for each indicator and the variability of these indicators between sites.

The article is organized as follows: Section 2 presents the measuring stations, their instrumentation and the check of the plausibility of the measurements as well as the McClear model. Section 3 describes the selection of clear-

sky conditions from measurements, the selection of periods of data from stations, and the methodology of comparisons between McClear estimates and ground-based measurements. The results of comparisons are given and discussed in Section 4. Possible explanations of the discrepancies between McClear and measurements are dealt with in Section 5. Section 6 includes a comparison between previous similar works and ours. Eventually, the conclusions are given in Section 7.



## 2 Data used

All data used in this research can be freely accessed through several public sources available on the Web. Details on access are given in Sect. 10 (Data availability).

### 2.1 Ground-based measurements

One-minute ground-based measurements of SSI, namely the global irradiance $G$, its diffuse component $D$ and its direct component $B$, or this component at normal incidence $B_N$, were collected from several networks offering more than 50 sites for periods ranging from 2010 to 2020, depending on the site. As explained later, the measurements were screened for their plausibility and only those made in clear-sky conditions were kept. Additional constraints on the minimal number of selected measurements during each year led to a restricted set of 44 stations. These stations are listed in Table 1 and a map is drawn in Fig. 1. Retained periods of measurements are discussed later. Table 2 lists the Köppen-Geiger climate type for each station according to Peel et al. (2007), while Table 3 lists the instruments used at each station.

The stations Gobabeb (Namibia) and De Aar (South Africa) belong to the Baseline Surface Radiation Network (BSRN, Ohmura et al., 1998) spread throughout the world. BSRN is a project of the World Climate Research Programme and maintains the highest standards in shortwave radiation measurements (Roesch et al., 2011; Vuilleumier et al., 2014). These stations are equipped with class A (formerly, secondary-standard) thermopile pyranometers, with one including a rotating shadowball, to measure separately $G$ and $D$ with a regular sampling of 1 min and pyrheliometers to measure $B_N$.

The South African Universities Radiation Network (SAURAN) use thermopile pyranometers and pyrheliometers similar to those in the BSRN network for measuring $G$, $D$, and $B_N$ every 1 min (Brooks et al., 2015). Nineteen stations in South Africa, one in Namibia, and one in Botswana are included in the study.

The remaining stations are financially supported by the World Bank Group apart four stations supported by the Maldivian corresponding local airports (Hanimaadhoo, Male, and Kadhdhoo), five stations supported by the Zambian Agricultural Research Institute (Kasama, Mutanda, Kaoma, Chilanga and Choma), one supported by the University of Zambia (Lusaka), one station supported by the Malawian University of Mzuzu (Mzuzu) and two stations supported by the Malawian Ministry of Natural Resources, Energy and Mining (Kasungu and Chileka). All are operated by companies. The instruments used at these stations are diverse and are reported in Table 3. All stations are equipped with at least one class A thermopile pyranometer which provides time-series of $G$. A few stations comprise two class A pyranometers for $G$. In these cases, we have arbitrarily selected the time series measured by the pyranometers labelled number 1 by the operator in the description of the station. It could have been possible to compare the two data sets but we had no means to decide which one to keep as we had not all necessary information on day-to-day operation on each instrument. This could be best done by the site operator. In other cases, stations are equipped with two instruments measuring $G$. In these cases, we have kept the data set acquired with the most precise instrument, e.g., class A instrument.

In most stations supported by the World Bank Group, the diffuse component $D$ is provided by either rotating shadowband irradiometers which measure almost simultaneously $G$ and $D$ on a horizontal plane every 1 min or Delta-T SPN1 pyranometers which comprise seven thermopiles and measure $G$ and $D$ simultaneously. In both cases, the direct component on a horizontal plane $B$ is deduced from $G$ and $D$ by the closure equation: $B=G-D$.



For the sake of the comparison with the other stations, $B$ is converted into $B_N$ by dividing $B$ by $\cos(\theta_S)$ where $\theta_S$ is the solar zenithal angle and is here computed with the SG2 algorithm (Blanc, Wald, 2012).


**Table 1. Description of measuring stations used for validation, ordered by decreasing latitude.**

| # | Station and country | Latitude (°) | Longitude (°) | Elevation (a.s.l. m) | Elevation in the CAMS cell (m) |
|---|---|---|---|---|---|
| 1 | Touba, Senegal | 14.77 | −15.92 | 37 | 27 |
| 2 | Fatick, Senegal | 14.37 | −16.41 | 8 | 22 |
| 3 | Kahone, Senegal | 14.17 | −16.03 | 10 | 24 |
| 4 | Hanimaadhoo, Maldives | 6.75 | 73.17 | 2 | 0 |
| 5 | Male, Maldives | 4.19 | 73.53 | −8 | 0 |
| 6 | Wadelai, Uganda | 2.73 | 31.39 | 644 | 969 |
| 7 | Kadhdhoo, Maldives | 1.86 | 73.52 | 0 | 0 |
| 8 | Laisamis, Kenya | 1.60 | 37.80 | 576 | 774 |
| 9 | Homa Bay, Kenya | −0.76 | 34.36 | 1335 | 1484 |
| 10 | Narok, Kenya | −1.32 | 35.71 | 1914 | 1714 |
| 11 | Shinyanga, Tanzania | −3.62 | 33.52 | 1179 | 1252 |
| 12 | Dodoma, Tanzania | −6.18 | 35.70 | 1139 | 1172 |
| 13 | Dar Es Salaam-TZ, Tanzania | −6.78 | 39.20 | −122 | 154 |
| 14 | Kasama, Zambia | −10.17 | 31.23 | 1379 | 1320 |
| 15 | Mzuzu, Malawi | −11.42 | 34.00 | 1285 | 974 |
| 16 | Mutanda, Zambia | −12.42 | 26.22 | 1316 | 1292 |
| 17 | Ndeke, Zambia | −12.58 | 28.29 | 1287 | 1221 |
| 18 | Kasungu, Malawi | −13.02 | 33.47 | 1065 | 960 |
| 19 | Kaoma, Zambia | −14.84 | 24.93 | 1170 | 1132 |
| 20 | Fig Tree, Zambia | −15.00 | 28.55 | 1143 | 1018 |
| 21 | Mumbwa, Zambia | −15.09 | 27.00 | 1103 | 1132 |
| 22 | Lusaka, Zambia | −15.39 | 28.34 | 1262 | 991 |
| 23 | Chilanga, Zambia | −15.55 | 28.25 | 1224 | 981 |
| 24 | Chileka, Malawi | −15.68 | 34.97 | 767 | 602 |
| 25 | Choma, Zambia | −16.84 | 27.07 | 1282 | 993 |
| 26 | Windhoek, Namibia | −22.57 | 17.08 | 1683 | 1420 |
| 27 | Vuwani, South Africa | −23.13 | 30.42 | 628 | 668 |
| 28 | Gobabeb, Namibia | −23.56 | 15.04 | 407 | 547 |
| 29 | Gaborone, Botswana | −24.66 | 25.93 | 1014 | 1124 |
| 30 | Pretoria- CSIR, South Africa | −25.75 | 28.28 | 1400 | 1358 |
| 31 | Pretoria- GIZ, South Africa | −25.75 | 28.23 | 1410 | 1355 |
| 32 | Witbank, South Africa | −25.89 | 29.12 | 1629 | 1375 |
| 33 | Alexander Bay, South Africa | −28.56 | 16.76 | 141 | 431 |
| 34 | Kwadlangezwa, South Africa | −28.85 | 31.85 | 90 | 391 |
| 35 | Bloemfontein-CUT, South Africa | −29.12 | 26.22 | 1397 | 1456 |
| 36 | Durban-KZW, South Africa | −29.82 | 30.94 | 200 | 558 |
| 37 | Durban-KZH, South Africa | −29.87 | 30.98 | 150 | 527 |
| 38 | De Aar, South Africa | −30.67 | 23.99 | 1287 | 1249 |
| 39 | Vanrhynsdorp, South Africa | −31.62 | 18.74 | 130 | 438 |
| 40 | Graaff-Reinet, South Africa | −32.49 | 24.59 | 660 | 928 |
| 41 | Alice, South Africa | −32.78 | 26.85 | 540 | 607 |
| 42 | Mariendal, South Africa | −33.85 | 18.82 | 178 | 284 |
| 43 | Stellenbosch, South Africa | −33.93 | 18.87 | 119 | 277 |
| 44 | Port Elizabeth, South Africa | −34.01 | 25.67 | 35 | 252 |



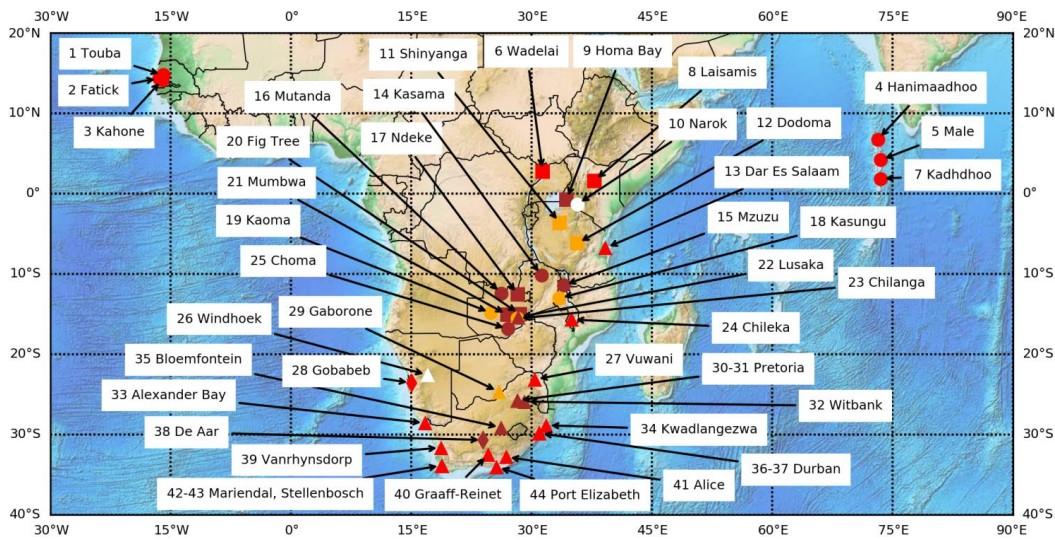

**Figure 1: Map of the stations. Diamonds are for the BSRN stations, triangles are for the other stations equipped with pyrheliometers. Circles and squares denote stations equipped with rotating shadowband irradiometers and with Delta-T SPN1 instruments respectively. Red color means an elevation less than 900 m. Orange means an elevation between 900 and 1200 m, brown an elevation between 1300 and 1600 m, and white an elevation greater than 1700 m. Numbers refer to the rank of the station in Table 1. The basemap is under public domain. The orographic basemap is from the Etopo1 data set from the National Oceanic and Atmospheric Administration of the United States of America.**

**Table 2. List of Köppen-Geiger climate types and corresponding stations, according to Peel et al. (2007).**

| Climate | Stations |
|---|---|
| Am: Tropical climate of monsoon type | Hanimaadhoo, Male, Kadhdhoo |
| Aw: Tropical climate of savannah type | Wadelai, Laisamis, Shinyanga, Dar Es Salaam, Chileka |
| BSh: Arid and hot climate of steppe type | Touba, Fatick, Kahone, Dodoma, Gaborone, Graaff-Reinet |
| BSk: Arid and cold climate of steppe type | Bloemfontein-CUT, De Aar, Vanrhynsdorp |
| BWh: Arid and hot climate of desert type | Windhoek, Gobabeb |
| BWk: Arid and cold climate of desert type | Alexander Bay |
| Cfa: Temperate climate without dry season and hot summer | KwaDlangezwa, Durban-KZW, Durban-KZH |
| Cfb: Temperate climate without dry season and warm summer | Alice, Port Elizabeth |
| Csa: Temperate climate with dry and hot summer | Mariendal, Stellenbosch |
| Cwa: Temperate climate with dry winter and hot summer | Kasama, Mzuzu, Mutanda, Ndeke, Kasungu, Kaoma, Fig Tree, Mumbwa, Lusaka, Chilanga, Choma, Vuwani |
| Cwb: Temperate climate with dry winter and warm summer | Homa Bay, Narok, Pretoria-CSIR, Pretoria-GIZ, Witbank |

**Table 3. Instruments used at each station.**

| Instrument | Stations |
|---|---|
| Class A thermopile pyranometers for $G$ and $D$, class A pyrheliometer for $B_N$. BSRN and SAURAN networks | Windhoek, Vuwani, Gobabeb, Gaborone, Pretoria-CSIR, Pretoria-GIZ , Witbank, Alexander Bay, KwaDlangezwa , Bloemfontein-CUT, Durban-KZW, Durban-KZH, De Aar, Vanrhynsdorp, Graaff-Reinet, Alice, Mariendal, Stellenbosch, Port Elizabeth |
| Class A thermopile pyranometers for $G$ and $D$, class A pyrheliometer for $B_N$ | Dar Es Salaam-TZ, Chileka |
| Class A thermopile pyranometers for $G$ and $D$, rotating shadowband irradiometer for $G$ and $D$, class A pyrheliometer for $B_N$ | Lusaka |





| Class A thermopile pyranometer and class B or C silicon pyranometer for *G*, rotating shadowband irradiometer for *D* | Touba, Fatick, Kahone |
|---|---|
| Class A thermopile pyranometer for *G*, rotating shadowband irradiometer for *G* and *D* | Hanimaadhoo, Male, Kadhdhoo, Kasama, Mzuzu, Mutanda, Kasungu, Kaoma, Chilanga, Choma |
| Class A thermopile pyranometers for *G*, Delta-T SPN1 pyranometer for *D* | Wadelai, Laisamis, Homa Bay, Narok, Shinyanga, Dodoma, Ndeke, Fig Tree, Mumbwa |

The World Meteorological Organization (WMO, 2018) sets recommendations for achieving a given accuracy in measuring solar radiation. Each element of a measurement system contributes to the final uncertainty of the data and the accuracy of solar radiation measurements made at ground stations depends on the radiometer

specifications, proper installation and maintenance, data acquisition method and accuracy, calibration method and frequency, location, environmental conditions, and possible real-time or a posteriori adjustments to the data (Sengupta et al., 2021). The WMO document clearly states that "good quality measurements are difficult to achieve in practice, and for routine operations, they can be achieved only with modern equipment and redundant measurements." In the WMO document, the typical relative uncertainty (95 % probability) of measurements of

good quality is 8 % for $G$ and $D$, and 2 % for $B_N$ with a minimum uncertainty of approximately 17 W m$^{-2}$ for the latter. The uncertainty targets are more stringent for BSRN measurements: 2 % for $G$ and $D$ and 0.5 % for $B_N$ (Ohmura et al., 1998). A very detailed analysis of the uncertainty of measurements made at the BSRN station of Payerne, Switzerland, was performed by Vuilleumier et al. (2014). They reported that the target can be achieved for $G$ and $D$ but not for $B_N$ for which the uncertainty is approximately 1.5 %. As for rotating shadowband

irradiometer, Wilbert et al. (2016) made a detailed analysis of the uncertainty of measurements acquired by a very well-maintained instrument at the PSA station in Spain. They found that the effects of the correction functions, including the spectral irradiance errors, are significant and wrote that uncertainties for corrected 1 min data are estimated to be 2.2 % for $G$ and 3.2 % for $B_N$ for both $G$ and $B_N$ greater than 300 W m$^{-2}$. In the more general case, Sengupta et al. (2021) report uncertainties of 4 % for $G$ and 5 % for $B_N$. As for the Delta-T SPN1

pyranometer, the uncertainty given by the manufacturer is 8 % for $G$ and $D$ with a minimum uncertainty of approximately 10 W m$^{-2}$. However, biases have been found as a function of $\theta_S$ and uncertainties may be greater than those given by the manufacturer (Badosa et al., 2014). Möllenkamp et al. (2020) found that monthly recalibration of the instrument against a pyrheliometer significantly reduces the uncertainty.

### 2.2 Checking the plausibility of the measurements

All the measurements have a temporal resolution of 1 min. Some were flagged by the station operators after a quality-check and only those flagged as non-suspicious were retained. We have performed an additional plausibility check on the data whose aim is not to question the quality flags provided by the station operators, but to check that we made no error in downloading the data and handling them. The tests used here originate from several articles; they are summarized in Korany et al. (2016) or Wald (2021) and reported here for the sake of

clarity. The plausibility tests check whether the measurements exceed physically possible and extremely rare limits as well as the consistency between the coincident measurements of $G$, $D$ and $B_N$. Let $E0_N$ and $E0$ denote the solar radiation impinging at the top of the atmosphere at normal incidence and on a horizontal surface respectively, with $E0=E0_N \cos(\theta_S)$. At any time and any site, $E0_N$ and $\theta_S$ are computed by the means of the SG2 algorithm (Blanc and Wald, 2012). The tests comprise several constants which are given here in W m$^{-2}$.

The tests based on physically possible limits are:



$$0.03 \, E0 \leq G \leq \min \left(1.2 \, E0_N, 1.5 \, E0_N (\cos(\theta_S))^{1.2} + 100\right) \tag{1}$$

$$0.03 \, E0 \leq D \leq \min \left(0.8 \, E0_N, 0.95 \, E0_N (\cos(\theta_S))^{1.2} + 50\right) \tag{2}$$

$$0 \leq B_N \leq E0_N \tag{3}$$

The tests based on extremely rare limits are:

$$0.03 \, E0 \leq G \leq 1.2 \, E0_N (\cos(\theta_S))^{1.2} + 50 \tag{4}$$

$$0.03 \, E0 \leq D \leq 0.75 \, E0_N (\cos(\theta_S))^{1.2} + 30 \tag{5}$$

$$0 \leq B_N \leq 0.95 \, E0_N (\cos(\theta_S))^{1.2} + 10 \tag{6}$$

The tests on consistency between independent measurements are only applied if $G > 50$ W m$^{-2}$. If $G$ and $D$ are given by two independent instruments, the test is:

$$D \leq 1.1 \, G \tag{7}$$

If $G$, $D$ and $B_N$ are given by three independent instruments, the test is:

$$0.92 \leq (D + B)/G \leq 1.08 \; if \; \theta_S \leq 75° \tag{8}$$

$$0.85 \leq (D + B)/G \leq 1.15 \; if \; \theta_S > 75°$$

Suspicious or erroneous measurements were flagged and then removed from the dataset. Then, a visual check was performed to detect and scrutinize outliers that are possibly rejected. In addition, we have put one more constraint on measurements. Since the lowest values can be noise and are therefore insignificant in a validation process, any measurement should be greater than a minimum significant value. If it was not, the measurement was removed from the dataset. The thresholds were selected in such a way such that there is a 99.7 % chance that the actual irradiances $G$, $D$ and $B_N$ are significantly different from 0 and that they can be used for the comparison. Based on the uncertainty of good quality measurements of $B_N$ as reported by the WMO (2018), the threshold was set to 1.5 times the minimum uncertainty, i.e., 26 W m$^{-2}$ for $B_N$. The WMO document does not give any minimum uncertainty for $G$ or $D$ and the thresholds were set arbitrarily to 30 W m$^{-2}$ for both.

### 2.3 The McClear model

The McClear model (Lefèvre et al., 2013; Gschwind et al., 2019) is built on abaci, also known as look-up-tables, by the means of the radiative transfer model libRadtran (Emde et al., 2016; Mayer and Kylling, 2005) based on the most improved Kato et al. (1999) approach (*katoandwandji*, Wandji Nyamsi et al., 2014; 2015). It accurately reproduces $G$, $D$, $B$ and $B_N$ computed by the libRadtran reference under clear-sky conditions with a computational speed approximately $10^5$ times greater (Lefèvre et al., 2013) thus offering the opportunity of delivering time-series of clear-sky irradiances at a given site within few seconds. To better exploit this advantage, and though it can be used as a standalone model, McClear is often exploited in combination with inputs from CAMS and other sources as a Web service freely delivering irradiances at the ground level for any period from 2004 until 2 days ago with different temporal summarizations (1 min, 15 min, 60 min, 1 day, 1 month) at any site in the world. In this sense, McClear is more an on-line service than a classic clear-sky model (Cros et al., 2013).

Besides the period of time, summarization and geographical location, usually provided by users, the McClear v3 service requires inputs that are automatically read from several sources. The total column contents in ozone and water vapor are given by CAMS as well as the total aerosol optical depth and the partial optical depth for sea salt, dust, organic matter, black carbon and sulfates aerosol species, all of them at 550 nm. Sea salt and dust aerosols have their sources linked to prognostic and diagnostic surface and near-surface model variables while





the organic matter, black carbon and sulphate aerosols have theirs read from external data sets. Removal processes include the dry deposition, including the turbulent transfer to the surface and the gravitational settling, and the wet deposition, including rainout and washout of aerosol particles in and below the clouds.

The readings from CAMS are resampled to the selected location by spatial bilinear interpolation and resampled in time to the desired summarization. Vertical profiles of temperature, pressure, density, and volume mixing ratio

for gases as a function of altitude are those from the AGFL (USA Air Force Geophysics Laboratory): tropics (afglt), mid-latitude summer and winter (afglmls and afglmlw), and sub-Arctic summer and winter (afglss and afglsw), as implemented in libRadtran. The reflective properties of the ground are calculated by the means of three parameters called *fiso*, *fvol*, and *fgeo* (Schaaf et al., 2002) that describe the bidirectional reflectance distribution functions (BRDF) and that are read from the twelve-monthly maps derived from the MODIS

datasets proposed by Blanc et al. (2014b). If not provided, the elevation of the site above mean sea level is taken from the Shuttle Radar Topography Mission datasets and the difference between this elevation or that provided by the user and the mean elevation in the CAMS cell is taken into account. The yearly average of $E0_N$ is known to astronomers as the total solar irradiance noted $E_{TSI}$. It is set to 1362 W m$^{-2}$ in McClear v3 as found by Meftah et al. (2014) in 2010. This is slightly less than the value 1367 W m$^{-2}$ adopted in McClear v2. At any time and

any site, $E0_N$, the solar zenithal angle $\theta_S$ and the solar azimuth are computed by the means of the SG2 algorithm (Blanc and Wald, 2012).

Version 2 was introduced in 2013 to partly palliate several discontinuities in space observed in outputs. A greater step was made with version 3 that removed several artifacts, including discontinuities in space and time in irradiance induced by non-linear algorithms for the selection of aerosol classes or vertical profiles from AGFL.

In addition, version 3 offers potentials for future improvement, especially regarding the description of aerosol properties. Finally, the replacement of the modified Beer-Lambert (MLB) function proposed by Mueller et al. (2004) by more complex functions describing the change of irradiance with $\theta_S$ combined with the computation of $B$ and $D$ instead of $G$ and $B$ allows an accurate computation of $D$ when the sun is below the horizon.

Comparisons were performed between McClear estimates and measurements made at eleven BSRN sites for

v1/v2 and v3 by Gschwind et al. (2019) who found similar results between the three versions. It follows that works dealing with the versions v1, v2 or v3 can be compared as we will do in Section 6.

McClear irradiances are freely accessible by machine-to-machine calls to the Web service McClear on the SoDa Service (Gschwind et al., 2006, www.soda-pro.com, last access: 2022-08-11) or manually through a Web interface. In the verbose mode, the flow returned by the service contains 1 min values of readings from CAMS

interpolated in space and time, namely, the optical depth of aerosols at 500 nm, and the total column contents in water vapor and ozone. It also contains 1 min values of $\theta_S$ calculated with the SG2 algorithm, and of irradiances at the top of atmosphere and at ground level, and ground albedo, calculated by McClear. This mode was conveniently exploited for the collection of McClear estimates for the same periods and same locations than the 1 min ground measurements.



**3 Screening of cloudless instants, selection of periods and methodology of validation**

**3.1 Screening of cloudless instants**

A screening algorithm needs to be applied on the ground measurements to separate the cloud-contaminated instants from the cloud-free ones. Several algorithms for detecting clear-sky instants from measurements have been published (see e.g. Bright et al., 2020; Calbó et al., 2001; Ellis et al., 2019; Long and Ackerman, 2000;

Reno and Hansen, 2016). Here, the algorithm of Lefèvre et al. (2013) was selected. Lefèvre et al. (2013) have found that the results of their algorithm provide less low values of SSI than that of Long and Ackerman (2000) and therefore wrote that their algorithm offers more confidence in the fact that the instant is clear. The possible influence of the algorithm for detecting clear-sky instants on results is discussed in Section 6.

Let $E0_N$ and $E0$ denote the solar radiation impinging at the top of the atmosphere at normal incidence and on a

horizontal surface respectively. The clearness index $KT$, direct normal clearness index $KT_{BN}$, and corrected clearness index $KT_{cor}$ (Ineichen and Perez, 1999) are respectively defined as:

$$KT = G/E0 \,, \tag{9}$$

$$KT_{BN} = B/E0 = B_N/E0_N \,, \tag{10}$$

$$KT_{cor} = KT/\left[1.031 \exp\left(-1.4 \middle/ \left(0.9 + 9.4/m\right)\right) + 0.1\right] \,, \tag{11}$$

where $m$ is the air mass defined by Kasten and Young (1989):

$$m(\theta_S) = \left(p/p_0\right)/\left[\cos(\theta_S) + 0.50572\,(\theta_S + 6.07995)^{-1.6364}\right] \,, \tag{12}$$

where $\theta_S$ is the solar zenithal angle expressed in degree, and $p$ and $p_0$ are respectively the pressure at the site under consideration and that at sea level. The ratio of pressures can be approximated as:

$$p/p_0 = \exp\left(-z/8435.2\right) \,, \tag{13}$$

where $z$ is the elevation above sea level expressed in m, and 8435.2 m is the scale height of the Rayleigh atmosphere. $KT$ is equal to the global transmissivity of the atmosphere, or atmospheric transmittance, or atmospheric transmission when the reflection of the ground is null. $KT_{cor}$ exhibits less dependence with $\theta_S$ than $KT$ (Ineichen and Perez, 1999).

The first filter in the Lefèvre et al. algorithm is a restriction on $D$ with respect to $G$ since $B$ is usually prominent

in cloud-free atmosphere:

$$D/G < 0.3 \,, \tag{14}$$

Only measurements that satisfied the first filter were retained. The second filter investigates the temporal fluctuation of the corrected clearness index $KT_{cor}$ since this amount must be stable for numerous hours in cloudless atmosphere. The first step of this filter is to retain only periods with enough measurements that have

passed the first filter. A given instant t, expressed in min, is kept only if at least 30 % of the 1 min observations in both intervals [t − 90 min, t] and [t, t + 90 min] have been retained after the first filter. An instant is considered clear if the standard deviation of $KT_{cor}$ in the interval [t − 90 min, t + 90 min] is less than a threshold, set empirically to 0.02. Only these 1 min clear-sky instants were retained for the validation. Because of the second filter, all retained instants are within [sunrise + 90 min, sunset – 90 min].



### 3.2 Selection of periods, number of samples and means of measurements

Apart the daily cycle of the solar zenithal angle, the SSI in cloudless conditions exhibits only one yearly noticeable cycle (Bengulescu et al., 2017, 2018) though the occurrence of cloud-free conditions varies with seasons. Hence, the year is an appropriate period for the validation of the McClear service. At a given station, a year is declared valid if the number of clear-sky instants is greater than a threshold arbitrarily set to 9 000. This threshold is large enough to account for various situations and is of the same magnitude than the mean number of measurements used at each station in Lefèvre et al. (2013). In addition, using yearly periods allows the analysis of changes in statistical indicators with time.

A few exceptions to the general rule were made in order to account for local conditions and to retain the greatest possible number of stations and measurements in the analysis. Namely, stations #1, 2, and 3 have only data from January to September in 2017. A pseudo-year 2017 was created at these stations, consisting of 12 consecutive months spanning over two years: 2016-10 / 2017-09. Similarly, a pseudo-year 2018 was created at the stations Ndeke (#17), Fig Tree (#20) and Mumbwa (#21) consisting of 12 consecutive months spanning from 2017-09 to 2018-08. Fig. 2 (left) gives the years retained at each station for $G$ and $B_N$ as well as the number of samples in each year. Table 5 reports the number of samples kept for validation at each station for all retained years. Most stations exhibit 1 or 2 years of data. Many SAURAN stations have more than 2 years of data. Stations Stellenbosch (#43), Gobabeb (#28) and Pretoria-GIZ (#31) have the longest records in this selection, respectively 9, 8, and 7 years.





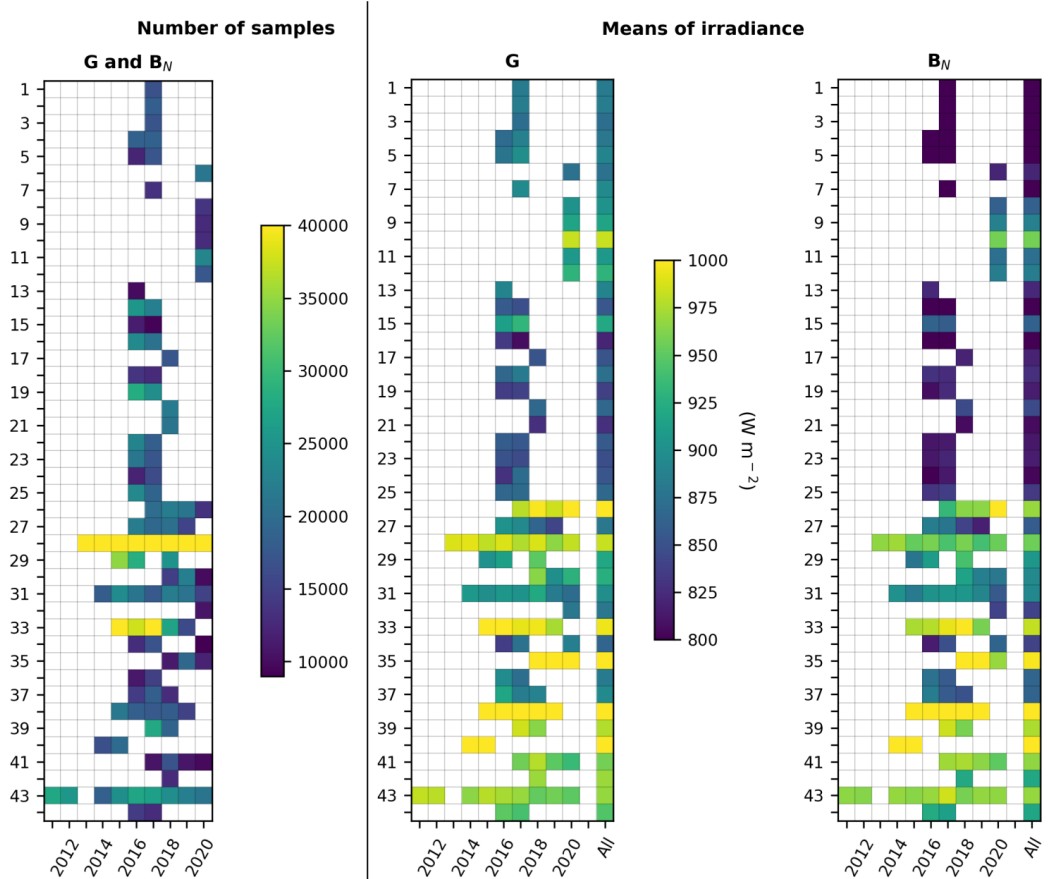

**Figure 2: Number of retained measurements (left) and means of $G$ and $B_N$ (right) at each station for each year. Also reported are the means for $G$ and $B_N$ for the whole period (All). Numbers refer to the rank of the station in Table 1.**

Fig. 2 (center and right) provides in a graphical form, the means of $G$ and $B_N$ for each year and for the whole period. The latter are also given in Table 5, as well as the means of $KT$ and $KT_{BN}$. As stations are ordered by decreasing latitudes, Fig. 2 shows an overall latitudinal trend in $G$, $B_N$, $KT$, and $KT_{BN}$ which tend to increase southwards. The trend is blurred by the different elevations, climates, total water vapor content, aerosol loading and possibly instrumentation for $B_N$. As a whole, the mean of $G$ is almost constant from station #1 to station #7, then increases with a local maximum at Narok (#10) likely because of its high elevation of 1914 m. Then, it exhibits a kind of trough at stations (#14-23, #25, #27) that experience the Cwa climate and show lower means of $G$ than the others though their elevation is often greater than 1000 m. The behavior is more confused at stations #26-44 though the mean tends to increase southwards with local maxima at Windhoek (#26), Bloemfontein-CUT (#35) and De Aar (#38) likely due to their elevation greater than 1300 m. Variables other than elevation intervene. For example, Pretoria (#30-31) or Witbank (#32) have elevations greater than 1400 m and exhibit lower means of $G$ than Alexander Bay (#33) or Graaff-Reinet (#40) whose elevations are respectively 141 m and 660 m. The three former experience the temperate Cwb climate while the two latter

 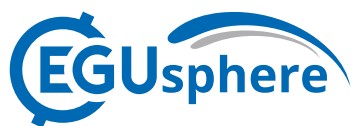

experience arid BWk and BSh climates. The minimum of $B_N$ is observed at Hanimadhoo (#4) and Male (#5). The mean of $B_N$ is often close to the mean of $G$ though less. Exceptions are observed at the stations in Senegal and Maldives where $B_N$ is much less than $G$. As $B_N$ and $G$ are close at the other stations, $B_N$ exhibits a behavior similar to that of $G$.

As a whole, means of $G$ and $B_N$ in cloudless conditions are fairly constant throughout years at a given station: changes from year to year are often less than 3 % of the yearly average, i.e. approximately 30 W m$^{-2}$. Magnitudes of changes are partly due to differences in atmospheric conditions, partly due to differences in the number of data available in each year and their distribution within the year. A special look at the very close stations, namely the two stations #30 (Pretoria-CSIR) and 31 (Pretoria-GIZ), less than 4 km apart having the

same elevation, and the stations Durban-KZW (#36) and Durban-KZH (#37) that are 7 km apart, with a difference in elevation of 50 m (200 m vs 150 m), reveals that the yearly mean of $B_N$ exhibits almost no difference between the closest stations in a given year. The differences are respectively 11, 5, and 23 W m$^{-2}$ in 2018, 2019 and 2020, at Pretoria, and respectively 10 and 1 W m$^{-2}$, in 2016 and 2017 at Durban. This is also true for $G$ at Durban: the differences are respectively 21 and 12 W m$^{-2}$ in 2016 and 2017. At Pretoria, the differences

in $G$ may be more pronounced depending on the year: 58, 17 and 59 W m$^{-2}$ in 2018, 2019 and 2020.

The mean clearness index $KT$ is comprised between 0.73 and 0.82 (Table 5). The greatest values are observed at elevated sites: Narok (#10, $KT$=0.80, 1914 m), Windhoek (#26, $KT$=0.81, 1683 m), Bloemfontein (#35, $KT$=0.82, 1397 m) and De Aar (#38, $KT$=0.82, 1287 m), with the exception of Graaff-Reinet (#40) which offers a mean $KT$ of 0.80 though its elevation is only 660 m. The clearness index $KT_{BN}$ is comprised between 0.54 and

0.76. Like for $KT$, the greatest values are found at elevated sites: Narok (#10, $KT_{BN}$=0.71), Windhoek (#26, $KT_{BN}$=0.71), Bloemfontein (#35, $KT_{BN}$=0.72) and De Aar (#38, $KT_{BN}$=0.76), with the exceptions of Alexander Bay (#33, $KT_{BN}$=0.71) and Graaff-Reinet (#40, $KT_{BN}$=0.73) of low elevation: 141 m and 660 m respectively.

**Table 5. Number of samples for validation, means s of $G$, $B_N$, $KT$, and $KT_{BN}$ at each station for the ensemble of retained years.**

| # | Station | Number of samples | Mean $G$ (W m$^{-2}$) | Mean $B_N$ (W m$^{-2}$) | Mean $KT$ | Mean $KT_{BN}$ |
|---|---|---|---|---|---|---|
| 1 | Touba | 16643 | 882 | 794 | 0.75 | 0.58 |
| 2 | Fatick | 18210 | 884 | 797 | 0.75 | 0.58 |
| 3 | Kahone | 17084 | 872 | 790 | 0.74 | 0.57 |
| 4 | Hanimaadhoo | 37259 | 879 | 751 | 0.74 | 0.55 |
| 5 | Male | 29399 | 890 | 750 | 0.75 | 0.54 |
| 6 | Wadelai | 21172 | 875 | 821 | 0.73 | 0.60 |
| 7 | Kadhdhoo | 13435 | 895 | 786 | 0.75 | 0.57 |
| 8 | Laisamis | 13837 | 902 | 861 | 0.76 | 0.64 |
| 9 | Homa Bay | 12695 | 917 | 887 | 0.76 | 0.65 |
| 10 | Narok | 12897 | 983 | 958 | 0.80 | 0.71 |
| 11 | Shinyanga | 23096 | 907 | 873 | 0.77 | 0.65 |
| 12 | Dodoma | 17419 | 930 | 884 | 0.77 | 0.65 |
| 13 | Dar Es Salaam-TZ | 9883 | 890 | 823 | 0.74 | 0.60 |
| 14 | Kasama | 47832 | 854 | 795 | 0.74 | 0.59 |
| 15 | Mzuzu | 20695 | 921 | 862 | 0.76 | 0.63 |
| 16 | Mutanda | 44324 | 821 | 782 | 0.73 | 0.59 |
| 17 | Ndeke | 17342 | 852 | 819 | 0.73 | 0.61 |
| 18 | Kasungu | 26341 | 874 | 827 | 0.75 | 0.61 |
| 19 | Kaoma | 52464 | 838 | 815 | 0.74 | 0.61 |
| 20 | Fig Tree | 21894 | 867 | 845 | 0.74 | 0.63 |
| 21 | Mumbwa | 20956 | 827 | 806 | 0.73 | 0.60 |
| 22 | Lusaka | 41389 | 857 | 813 | 0.74 | 0.60 |
| 23 | Chilanga | 38463 | 848 | 816 | 0.75 | 0.61 |
| 24 | Chileka | 28205 | 853 | 803 | 0.73 | 0.59 |





| 25 | Choma | 42667 | 867 | 834 | 0.76 | 0.62 |
|----|-------|-------|-----|-----|------|------|
| 26 | Windhoek | 76664 | 1003 | 969 | 0.81 | 0.71 |
| 27 | Vuwani | 77020 | 879 | 858 | 0.76 | 0.63 |
| 28 | Gobabeb | 399252 | 983 | 956 | 0.79 | 0.69 |
| 29 | Gaborone | 88168 | 919 | 906 | 0.76 | 0.66 |
| 30 | Pretoria- CSIR | 46943 | 926 | 894 | 0.78 | 0.65 |
| 31 | Pretoria- GIZ | 138636 | 899 | 893 | 0.76 | 0.65 |
| 32 | Witbank | 10625 | 879 | 840 | 0.75 | 0.61 |
| 33 | Alexander Bay | 164366 | 995 | 982 | 0.79 | 0.71 |
| 34 | Kwadlangezwa | 39000 | 865 | 841 | 0.75 | 0.61 |
| 35 | Bloemfontein-CUT | 42713 | 1031 | 998 | 0.82 | 0.72 |
| 36 | Durban-KZW | 26023 | 883 | 865 | 0.75 | 0.63 |
| 37 | Durban-KZH | 45825 | 896 | 864 | 0.76 | 0.63 |
| 38 | De Aar | 90684 | 1056 | 1051 | 0.82 | 0.76 |
| 39 | Vanrhynsdorp | 46627 | 977 | 975 | 0.79 | 0.70 |
| 40 | Graaff-Reinet | 36457 | 1009 | 1018 | 0.80 | 0.73 |
| 41 | Alice | 48409 | 958 | 967 | 0.78 | 0.70 |
| 42 | Mariendal | 12870 | 970 | 920 | 0.78 | 0.66 |
| 43 | Stellenbosch | 217291 | 968 | 968 | 0.78 | 0.69 |
| 44 | Port Elizabeth | 27495 | 946 | 918 | 0.78 | 0.66 |

### 3.3 Methodology of validation

The validation was performed by computing the differences between the McClear estimates and the measurements for coincident instants and location, for each year and for the ensemble of years. The differences

were summarized by their mean, known as the bias or mean bias error, their standard deviation, and the root mean square error. Relative values were expressed with respect to the means of the measurements for the corresponding period. The Pearson correlation coefficients, slopes and offsets of the least-squares fitting lines were computed as well as the ratios of estimates to measurements and the ratios of variances of estimates to those of measurements. Several graphs were also drawn such as 2D histograms of measurements and estimates,

histograms of each data set and histograms of differences as well as boxplots of ratio and differences as function of $\theta_S$, total column contents in ozone and water vapor and optical depth of aerosols at 550 nm.

These operations were performed for $G$, $B_N$, $KT$, and $KT_{BN}$ at each station for the whole data set, and also for subsets of data built for different years, different classes of $\theta_S$, different classes of readings from CAMS, namely optical depth of aerosols at 500 nm, and total column contents in water vapor and ozone, and different classes of

ground albedo read from McClear outputs, and graphs were drawn to assess the influence of these quantities. Results were analyzed as a function of the station, latitude, climate, elevation, year, means of irradiance or clearness index, variances of the measurements, instruments, and even operators, to evidence possible trends.

Since correlation coefficients close to 1 mean that measurements and estimates vary similarly in time and slopes close to 1 mean that the amplitudes of the variations are similar, expectations are correlation coefficients and

slopes of the fitting lines both close to 1. Biases are expected to be close to 0 and ratios of variances are expected to be close to 1. A bias greater than 0, respectively a ratio of variances greater than 1, would mean an overestimation of the mean and variance. Knowing that the relative standard deviation is half the relative uncertainty, the expectations about the relative standard deviation depend on the relative uncertainty of the measurements and are based on the hypothesis that the McClear outputs, excluding possible bias, meet the good

quality standard of the World Meteorological Organization (WMO, 2018). Using a bulk approach, we have assumed that the square of the uncertainties in this validation is the quadratic sum of the uncertainties of the



McClear estimates and of the uncertainties of the measurements (see e. g., Sengupta et al., 2021, for more complex approaches). The relative standard deviation of the errors in this validation, σ, is given by:

$$\sigma^2 = \sigma^2_{instrument} + \sigma^2_{McClear} ,$$ (15)

where $\sigma_{instrument}$ is half the relative uncertainty of the instrument and $\sigma_{McClear}$ is half that of the McClear outputs. $\sigma_{McClear}$ is hypothetically set to 4 % for $G$ and 1 % for $B_N$ as discussed in Section 2.1. At BSRN stations, $\sigma_{instrument}$ is set to 1 % for $G$ and 1 % for $B_N$. At other stations, $\sigma_{instrument}$ is set to 4 % for $G$. As for $B_N$, $\sigma_{instrument}$ is set to 1 % at stations equipped with pyrheliometers, to 2.5 % at stations equipped with rotating shadowband irradiometers, and to 4 % at stations equipped with SPN1 instruments. Table 6 provides the expectations for σ

depending on the instruments. σ should be less than or equal to the corresponding limits in Table 6 if the McClear outputs, excluding bias, meet the good quality standard of the World Meteorological Organization.

**Table 6. Expectations for the relative standard deviation of errors σ (%).**

|       | BSRN | Other pyranometers | Other pyrheliometers | Rotating shadowband irradiometers | SPN1 |
|-------|------|--------------------|----------------------|-----------------------------------|------|
| $G$   | 4.1  | 5.7                | -                    | -                                 | -    |
| $B_N$ | 1.4  | -                  | 1.4                  | 2.7                               | 4.1  |

**4 Results and discussion**

Figure 3 exhibits examples of the 2D histograms for $G$ and $B_N$ at the BSRN station Gobabeb (#28). At this station, the measurements of $G$ are well reproduced by the McClear service (left graph). The cloud of points is well elongated along the identity line with a very limited scattering. The scattering of points is more pronounced for $B_N$ (right graph). Other 2D histograms at each station are available as supplementary material as well as many

other graphs, including for clearness indices $KT$ and $KT_{BN}$.

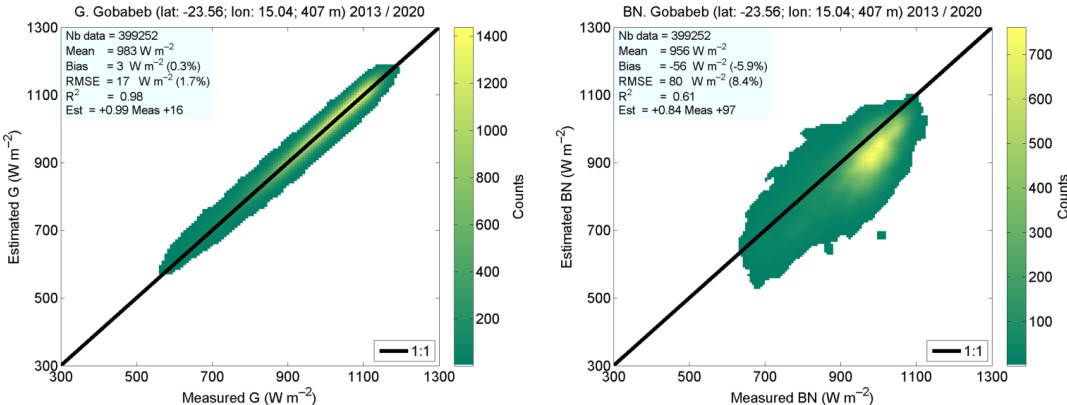

**Figure 3: 2D histograms between measurements (horizontal axis) and McClear-v3 estimates (vertical axis) for $G$ (left) and $B_N$ (right) at the BSRN station Gobabeb (#28). The color indicates the number of pairs in each class.**





### 4.1 Means and variances for *G*

Table 7 reports the bias, the relative bias and the ratio of variances at each station for *G*. The leftmost graphs in Figs. 4 and 5 exhibit respectively the bias and relative bias for *G* at each station for each year and for the whole period. The bias ranges between −27 W m$^{-2}$ at Male (#5, −3.1 % in relative value) and 57 W m$^{-2}$ at Witbank (#32, 6.5 %) (Table 7). If these extremes are removed, the bias lies within [−20, 45] W m$^{-2}$, i.e. [−2.3, 5.5] %, with a mean of 16 W m$^{-2}$ (1.8 %). The bias is most often positive (Fig. 4) and half of the stations exhibit small bias in the range [−15, 15] W m$^{-2}$. The bias shows a strong dependency with climate and with station to a lesser extent (Table 7 combined with Table 2). The bias is large and negative at the three stations in Maldives Archipelago experiencing a tropical climate of monsoon type (Am): −20, −27, and −20 W m$^{-2}$ (Table 7). On the contrary, stations in the tropical climate of savannah type (Aw, #6, 8, 11, 13, 24) offer positive bias, from 10 to 18 W m$^{-2}$, at the exception of the southernmost one (Chileka, #24, 40 W m$^{-2}$). The bias in arid climates (BSh, BSk, BWh, BWk) ranges between −7 W m$^{-2}$ and 15 W m$^{-2}$, with the exceptions of the elevated station #12 (19 W m$^{-2}$) in the BSh climate and the two southernmost stations #29 (37 W m$^{-2}$) and #40 (17 W m$^{-2}$). The stations in temperate climate without dry season (Cfa, Cfb, #34, 36-37, 41, 44) or with dry and hot summer (Csa, #42-43) exhibit biases from −8 W m$^{-2}$ to 16 W m$^{-2}$. Those in temperate climates with dry winter and hot summer (Cwa: #14-23, 25, and 27) exhibit the greatest biases in the interval [23, 45] W m$^{-2}$, i.e. [2.7, 5.5] %. Stations in temperate climates with dry winter and warm summer (Cwb, #9-10, 30-32) exhibit high variability in bias, from 5 W m$^{-2}$ (0.5 %) to 57 W m$^{-2}$ (6.5 %). Both BSRN stations (#28 Gobabeb, BWh, and #38 De Aar, BSk) exhibit a bias of only 3 W m$^{-2}$ (Table 7). Changes in bias from year to year at a given station are less than 10 W m$^{-2}$ in absolute value (approx. 1 % in relative value) in 84 % of cases (Fig. 4, left). Relative differences slightly greater than 1 % are observed for the same year at couples of close stations Pretoria (#30-31) and Durban (#36-37) (Fig. 5, left).

The ratio of variances ranges between 0.82 (#3, Kahone) and 1.15 (#42, Mariendal) (Table 7). If these extremes are excluded, the ratio lies in the interval [0.83, 1.05] with a mean of 0.95. It is often less than 1, which means an underestimation of the variance. About half of the stations (45 %) exhibit ratios comprised between 0.95 and 1.05 (Table 7). The link with elevation is unclear though stations of elevation greater than 1300 m have ratios less than 0.95 whatever the climate. There is a clear link with climates (Table 7 combined with Table 2). The ratios are often less than 0.95 in tropical climates (Am, Aw) and arid climates of steppe type (BSh, BSk). They are closer to 1 in arid climates of desert type (BWh, BWk) and temperate climates (Cfa, Cfb, Csa, Cwa). An exception is the temperate climate Cwb whose stations are above 1300 m and ratios are less than 0.93. Changes in ratio of variances from year to year at a given station range from −0.28 to 0.30 but are within the interval [−0.05, 0.05] in 63 % of the cases (not shown). Changes are less than 0.05 in absolute value at couples of close stations Pretoria (#30-31) and Durban (#36-37) for the same year.

In summary, there are links between elevation and climates and bias and ratio of variances. The variance is underestimated at elevation greater than 1300 m whatever the climate. In tropical climates, the variance of *G* is underestimated while the mean is underestimated (Am) or overestimated (Aw). In arid climates, McClear fairly correctly estimates the mean. It fairly correctly estimates the variance in climates of desert type (BWh, BWk) and underestimates it in climates of steppe type (BSh, BSk). In temperate climates, the variance is correctly estimated with the exception of stations in Cwb climate due to their high elevation. The mean is fairly correctly estimated in Cfa, Cfb, and Csa, and is strongly overestimated in Cwa and Cwb.




**Table 7. Bias (in W m$^{-2}$), relative bias (in %) and ratio of variances for $G$ at each station.**

| St. | Bias | Rel. bias | Ratio var. | St. | Bias | Rel. bias | Ratio var. | St. | Bias | Rel. bias | Ratio var. |
|---|---|---|---|---|---|---|---|---|---|---|---|
| 1 | 4 | 0.4 | 0.86 | 16 | 34 | 4.1 | 1.03 | 31 | 27 | 3.0 | 0.92 |
| 2 | −2 | −0.2 | 0.88 | 17 | 35 | 4.1 | 1.01 | 32 | 57 | 6.5 | 0.84 |
| 3 | 3 | 0.3 | 0.82 | 18 | 33 | 3.8 | 0.92 | 33 | 14 | 1.4 | 1.01 |
| 4 | −20 | −2.3 | 0.94 | 19 | 27 | 3.3 | 1.02 | 34 | 16 | 1.8 | 0.95 |
| 5 | −27 | −3.1 | 0.92 | 20 | 32 | 3.7 | 0.98 | 35 | −7 | −0.7 | 0.94 |
| 6 | 18 | 2.1 | 0.94 | 21 | 45 | 5.5 | 0.99 | 36 | 8 | 1.0 | 0.95 |
| 7 | −20 | −2.3 | 1.00 | 22 | 39 | 4.6 | 1.00 | 37 | −1 | −0.2 | 0.98 |
| 8 | 14 | 1.5 | 1.05 | 23 | 36 | 4.2 | 0.97 | 38 | 3 | 0.3 | 0.91 |
| 9 | 5 | 0.5 | 0.87 | 24 | 40 | 4.6 | 0.88 | 39 | 15 | 1.5 | 1.04 |
| 10 | 6 | 0.6 | 0.90 | 25 | 32 | 3.7 | 0.95 | 40 | 17 | 1.7 | 0.94 |
| 11 | 11 | 1.2 | 0.89 | 26 | 8 | 0.8 | 0.83 | 41 | 10 | 1.0 | 0.98 |
| 12 | 19 | 2.0 | 0.91 | 27 | 23 | 2.7 | 0.89 | 42 | 12 | 1.2 | 1.15 |
| 13 | 10 | 1.1 | 0.93 | 28 | 3 | 0.3 | 1.00 | 43 | 14 | 1.5 | 1.02 |
| 14 | 29 | 3.4 | 0.92 | 29 | 37 | 4.1 | 0.97 | 44 | −8 | −0.8 | 0.99 |
| 15 | 37 | 4.1 | 0.88 | 30 | 14 | 1.5 | 0.90 | | | | |



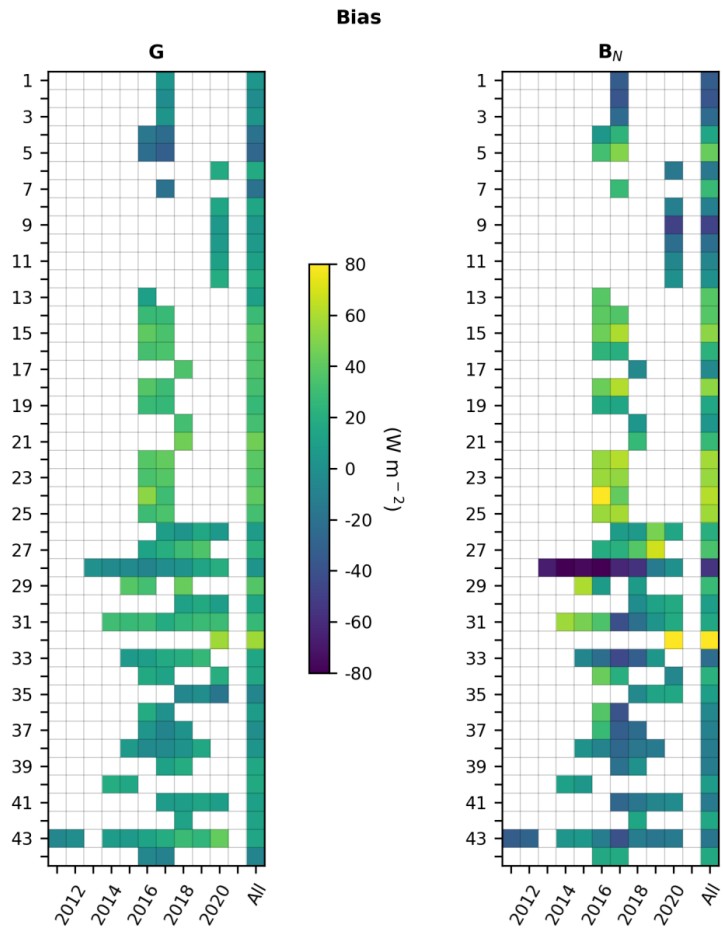

**Figure 4: Bias for *G* and *B_N* at each station for each year, and for the whole period (All). Numbers refer to the rank of the station in Table 1.**





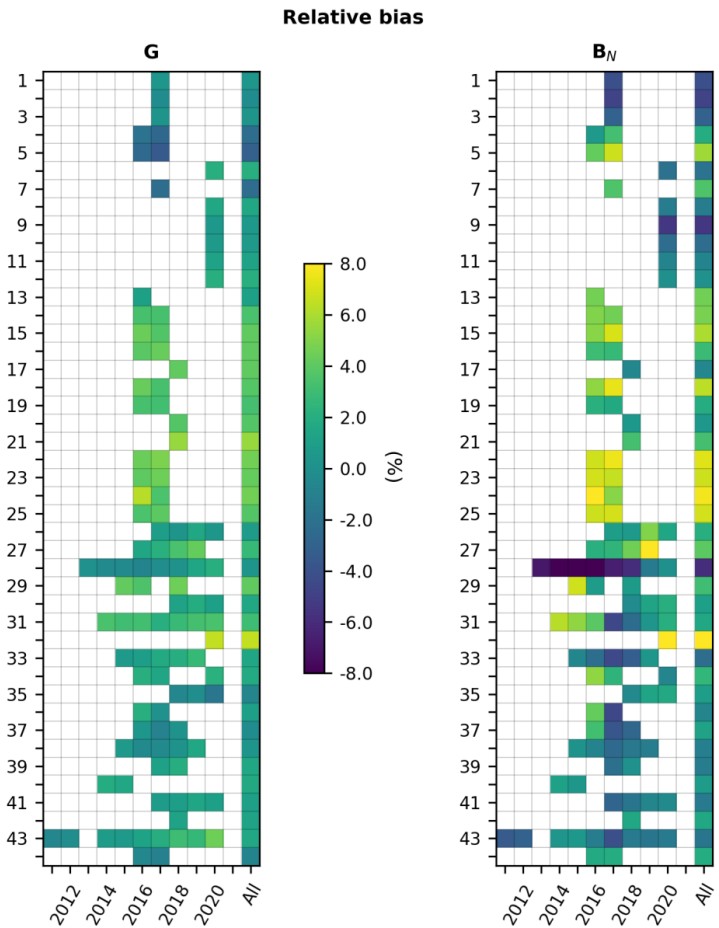

**Figure 5: Relative bias for *G* and *B_N* at each station for each year, and for the whole period (All). Numbers refer to the rank of the station in Table 1.**


## 4.2 Means and variances for $B_N$

Table 8 reports the bias, the relative bias and the ratio of variances at each station for $B_N$. The rightmost graphs in Figs. 4 and 5 exhibit respectively the bias and relative bias for $B_N$ at each station for each year and for the whole period. The bias ranges between −56 W m⁻² at Gobabeb (#28, −5.9 % in relative value) and 90 W m⁻² at

Witbank (#32, 10.7 %) (Table 8). If these two extremes are removed, the bias lies in a narrower interval [−48, 62] W m⁻², i.e. [−5.4, 7.7] %, with a mean of 11 W m⁻² (1.4 %). The bias is positive at 27 stations (64 % of the total) (Table 8) and 13 stations exhibit small relative bias in the range [−1.5, 1.5] %, i.e. 29 % of the stations. The influence of the climate or elevation is much less marked than for *G* for both the bias and ratio of variance. One may note that stations located in climates Am and Cwa exhibit positive biases among the greatest.

Changes in bias from year to year at a given station are less than 10 W m⁻² in absolute value in 40 % of cases (approx. 1 % in relative value) (Fig. 4, right). Relative differences around 1 % are observed for the same year at



couples of close stations Pretoria (#30-31) and Durban (#36-37) (Fig. 5 right). The widths of the intervals [−48, 62] W m$^{-2}$ for the bias throughout the stations and [−5.4, 7.7] % for the relative bias are 110 W m$^{-2}$ and 13.1 % respectively (Table 8): there is a large variation of the bias within the set of stations.

The ratio of variances ranges between 0.48 (#32, Witbank) and 2.29 (#42, Mariendal) (Table 8). If these extremes are excluded, the ratio lies in the interval [0.65, 1.50] with a mean of 1.07. The ratio exceeds 1 at 24 stations (overestimation) and is less than 1 at the 20 others. Nine stations exhibit ratios comprised between 0.95 and 1.05. Changes in ratio of variances from year to year at a given station range from −0.61 to 0.98 and are within the interval [−0.10, 0.10] in 20 % of the cases only (not shown). Changes are less than 0.1 in absolute

value at couples of close stations Pretoria (#30-31) and Durban (#36-37) for the same year. The width of the interval of variations in ratio throughout the stations is 0.85, i.e., 79.1 % relative to the middle of the range (Table 8). It can be concluded that McClear tends to overestimate both the means and variances of $B_N$ though these conclusions depend on the stations as the bias and the ratio of variances can hardly be considered as constant in time and space.


**Table 8. Bias (in W m$^{-2}$), relative bias (in %) and ratio of variances for $B_N$ at each station.**

| St. | Bias | Rel. bias | Ratio var. | St. | Bias | Rel. bias | Ratio var. | St. | Bias | Rel. bias | Ratio var. |
|---|---|---|---|---|---|---|---|---|---|---|---|
| 1 | −33 | −4.1 | 1.42 | 16 | 23 | 2.9 | 1.47 | 31 | 15 | 1.6 | 0.98 |
| 2 | −37 | −4.7 | 1.45 | 17 | −5 | −0.6 | 1.07 | 32 | 90 | 10.7 | 0.48 |
| 3 | −24 | −3.0 | 1.50 | 18 | 53 | 6.4 | 0.97 | 33 | −23 | −2.3 | 1.12 |
| 4 | 14 | 1.9 | 1.50 | 19 | 16 | 1.9 | 1.35 | 34 | 22 | 2.6 | 0.93 |
| 5 | 43 | 5.7 | 1.07 | 20 | 4 | 0.5 | 1.04 | 35 | 9 | 0.9 | 0.69 |
| 6 | −16 | −1.9 | 1.13 | 21 | 27 | 3.3 | 1.03 | 36 | −6 | −0.7 | 1.48 |
| 7 | 28 | 3.6 | 0.86 | 22 | 58 | 7.2 | 0.94 | 37 | −10 | −1.2 | 1.38 |
| 8 | −11 | −1.3 | 1.46 | 23 | 55 | 6.8 | 1.12 | 38 | −12 | −1.1 | 1.08 |
| 9 | −48 | −5.4 | 0.82 | 24 | 62 | 7.7 | 0.73 | 39 | −12 | −1.2 | 1.27 |
| 10 | −22 | −2.3 | 0.76 | 25 | 57 | 6.9 | 1.02 | 40 | 8 | 0.8 | 0.99 |
| 11 | −7 | −0.8 | 0.83 | 26 | 20 | 2.0 | 0.68 | 41 | −14 | −1.5 | 0.96 |
| 12 | 0 | 0.1 | 0.82 | 27 | 34 | 4.0 | 0.83 | 42 | 15 | 1.6 | 2.29 |
| 13 | 38 | 4.6 | 0.65 | 28 | −56 | −5.9 | 1.16 | 43 | −18 | −1.8 | 1.36 |
| 14 | 38 | 4.8 | 1.21 | 29 | 28 | 3.1 | 0.97 | 44 | 17 | 1.9 | 1.05 |
| 15 | 52 | 6.0 | 0.78 | 30 | 9 | 1.0 | 0.83 | | | | |

**4.3 Correlation coefficients and slopes of the least-squares fitting lines for $G$**

Figure 6 (left part) exhibits the correlation coefficients for $G$ at each station for each year and for the whole

period while Table 9 reports the correlation coefficients and slopes of the least-squares fitting lines at each station. The McClear estimates for $G$ correlate very well with the measurements at all stations. This was expected because of the strong influence of the solar zenithal angle. Correlation coefficients are comprised between 0.960 and 0.991 (Table 9) and are greater than 0.970 at 35 stations (80 % of the stations). The greater the mean $KT$, the greater the correlation coefficient (Table 9 combined with Table 5). Slopes are comprised

between 0.88 (#3, Kahone) and 1.05 (#42, Mariendal) and are comprised between 0.95 and 1.05 at 25 stations (57 % of the stations) (Table 9). Correlation coefficients vary very little throughout years at a given station: changes are less than 0.02 (Fig. 6, left). In addition, they are very close to each other at each station of both couples of close stations Pretoria (#30-31) and Durban (#36-37) for the same years (Fig. 6, left). This is also true for the slopes with relative changes from year to year less than 10 %. The correlation coefficients, respectively

the slopes, are very close to each other within the same climatic area whatever the elevation of the stations





(Table 9 combined with Table 2). It can be concluded that the correlation coefficients and the slopes have little variation in time and can be considered as constant in time and at close locations.

The width of the interval [0.960, 0.991] of the correlation coefficients throughout the stations is 0.031, i.e., 3 % relative to the middle of the range (Table 9). This is very small: it can be reasonably assumed that the correlation

coefficient varies very little in space. If the minimum in slope (0.88 at #3, Kahone) is excluded, the slopes lie in the interval [0.90, 1.05] whose width is 0.15, i.e., 15 % relative to the middle of the range (Table 9), which means that the spatial variation in slopes is noticeable at mesoscales. We conclude that the minute-to-minute variability in $G$ is well reproduced by McClear at all stations though there is a tendency at some places to overestimate the smallest irradiances and underestimate the greatest ones, with unpredictable variations of this

tendency in space.

**Table 9. Correlation coefficients (Correl. coeff.) and slopes of the least-squares fitting lines at each station for *G*.**

| St. | Correl. coeff. | Slope | St. | Correl. coeff. | Slope | St. | Correl. coeff. | Slope |
|---|---|---|---|---|---|---|---|---|
| 1 | 0.975 | 0.90 | 16 | 0.960 | 0.97 | 31 | 0.986 | 0.94 |
| 2 | 0.979 | 0.92 | 17 | 0.964 | 0.97 | 32 | 0.983 | 0.90 |
| 3 | 0.967 | 0.88 | 18 | 0.973 | 0.93 | 33 | 0.984 | 0.99 |
| 4 | 0.977 | 0.95 | 19 | 0.971 | 0.98 | 34 | 0.991 | 0.96 |
| 5 | 0.977 | 0.94 | 20 | 0.969 | 0.96 | 35 | 0.983 | 0.95 |
| 6 | 0.968 | 0.94 | 21 | 0.968 | 0.96 | 36 | 0.987 | 0.96 |
| 7 | 0.985 | 0.98 | 22 | 0.974 | 0.97 | 37 | 0.991 | 0.98 |
| 8 | 0.987 | 1.01 | 23 | 0.975 | 0.96 | 38 | 0.978 | 0.93 |
| 9 | 0.975 | 0.91 | 24 | 0.965 | 0.91 | 39 | 0.986 | 1.01 |
| 10 | 0.979 | 0.93 | 25 | 0.977 | 0.95 | 40 | 0.989 | 0.96 |
| 11 | 0.975 | 0.92 | 26 | 0.990 | 0.90 | 41 | 0.991 | 0.98 |
| 12 | 0.980 | 0.93 | 27 | 0.986 | 0.93 | 42 | 0.978 | 1.05 |
| 13 | 0.983 | 0.95 | 28 | 0.989 | 0.99 | 43 | 0.966 | 0.98 |
| 14 | 0.963 | 0.92 | 29 | 0.983 | 0.97 | 44 | 0.990 | 0.98 |
| 15 | 0.965 | 0.91 | 30 | 0.991 | 0.94 |  |  |  |



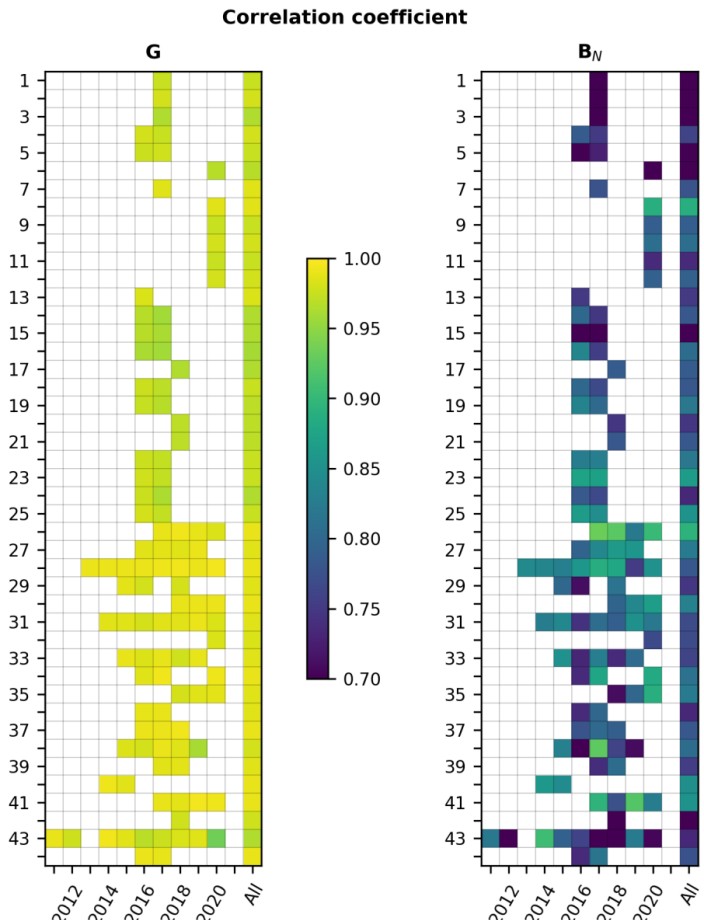

**Figure 6: Correlation coefficient for *G* and *B_N* at each station for each year, and for the whole period (All). Numbers refer to the rank of the station in Table 1.**

## 4.4 Correlation coefficients and slopes of the least-squares fitting lines for $B_N$

Figure 6 (right part) exhibits the correlation coefficients for $B_N$ at each station for each year and for the whole period. Table 10 reports the correlation coefficients and slopes of the least-squares fitting lines at each station for $B_N$. The McClear estimates for $B_N$ correlate well with the measurements at most stations. The correlation coefficient for $B_N$ ranges between a minimum of 0.532 observed at Mariendal (#42) and a maximum of 0.896 at Windhoek (#26) (Table 10). It is greater than or equal to 0.700 at all stations, except six: #1, 3, 5, 6, 15, and 42. Similarly to *G*, the greater the mean $KT_{BN}$, the greater the correlation coefficient (Tables 5 and 10). Slopes are less than 1 except at Laisamis (#8) and are often close to 0.80 with a minimum of 0.53 at Witbank (#32) (Table 10). Correlation coefficients for $B_N$ may vary or not with years at a given station: changes greater than 0.1 may be observed at stations #28-29, 31, 34-35, 38, 41 and 43 (Fig. 6, right). Changes in slope may also be highly variable depending on the station and the year. Correlation coefficients, respectively slopes, are constant at short





spatial scales: they are very close to each other at each station of both couples of close stations Pretoria (#30-31)
and Durban (#36-37) for the same years (Fig. 6, right).

The correlation coefficients and slopes vary significantly from one station to the other (Table 10). If the six
stations #1, 3, 5, 6, 15, and 42 are excluded, the correlation coefficient for $B_N$ lies in the interval [0.700, 0.896],
whose width is 0.196, i.e. 25 % relative to the middle of the range. The change in space in correlation coefficient
is important and cannot be neglected. If the minimum in slope (0.53 at #32, Witbank) is excluded, the slopes lie
in the interval [0.58, 1.07] whose width is 0.49, i.e. 60 % relative to the middle of the range, which is important.
If the four smallest slopes are excluded (stations #13, 15, 24, and 32), the slopes vary between 0.67 and 1.07.
The width of the range is 0.40, i.e. 46 % relative to the middle of the range, which is still large. We conclude that
the minute-to-minute variability in $B_N$ is fairly well reproduced by McClear at all stations though there is an
overestimation of the smallest irradiances and an underestimation of the greatest ones, the magnitudes of these
being unpredictable in time and space.

**Table 10. Correlation coefficients (Correl. coeff.) and slopes of the least-squares fitting lines at each station for $B_N$.**

| St. | Correl. coeff. | Slope | St. | Correl. coeff. | Slope | St. | Correl. coeff. | Slope |
|---|---|---|---|---|---|---|---|---|
| 1 | 0.676 | 0.81 | 16 | 0.803 | 0.98 | 31 | 0.768 | 0.76 |
| 2 | 0.700 | 0.84 | 17 | 0.790 | 0.82 | 32 | 0.768 | 0.53 |
| 3 | 0.635 | 0.78 | 18 | 0.779 | 0.77 | 33 | 0.764 | 0.81 |
| 4 | 0.762 | 0.93 | 19 | 0.821 | 0.95 | 34 | 0.813 | 0.78 |
| 5 | 0.698 | 0.72 | 20 | 0.746 | 0.76 | 35 | 0.825 | 0.69 |
| 6 | 0.658 | 0.70 | 21 | 0.784 | 0.79 | 36 | 0.735 | 0.89 |
| 7 | 0.777 | 0.72 | 22 | 0.821 | 0.80 | 37 | 0.781 | 0.92 |
| 8 | 0.888 | 1.07 | 23 | 0.864 | 0.92 | 38 | 0.807 | 0.84 |
| 9 | 0.790 | 0.71 | 24 | 0.737 | 0.63 | 39 | 0.755 | 0.85 |
| 10 | 0.808 | 0.71 | 25 | 0.856 | 0.87 | 40 | 0.851 | 0.85 |
| 11 | 0.736 | 0.67 | 26 | 0.896 | 0.74 | 41 | 0.852 | 0.83 |
| 12 | 0.792 | 0.72 | 27 | 0.825 | 0.75 | 42 | 0.532 | 0.81 |
| 13 | 0.751 | 0.61 | 28 | 0.780 | 0.84 | 43 | 0.737 | 0.86 |
| 14 | 0.778 | 0.85 | 29 | 0.751 | 0.74 | 44 | 0.773 | 0.79 |
| 15 | 0.656 | 0.58 | 30 | 0.830 | 0.76 | | | |

### 4.5 Standard deviations of errors and RMSE for $G$

Table 11 reports the standard deviation of errors σ, the RMSE and their values relative to the mean of the
measurements at each station for $G$. The leftmost graphs in Figs. 7 and 8 exhibit respectively the standard
deviation of errors and its relative value for $G$ at each station for each year and for the whole period. The
standard deviation ranges between 13 W m$^{-2}$ (1.3 %) at Graaff-Reinet (#40) and 31 W m$^{-2}$ (3.7%) at Chileka
(#24), which is a very limited range (Table 11). If these extremes are removed, the standard deviation lies within
[13, 30] W m$^{-2}$, with a mean of 22 W m$^{-2}$ (2.4 %). This is small and 37 stations out of 44 report a relative
standard deviation less than 3 %. The RMSE for $G$ ranges between 16 W m$^{-2}$ (1.7 %) at station #41 (Alice) and
#44 (Port Elizabeth) and 63 W m$^{-2}$ (7.1 %) at station #32 (Witbank) (Table 11). If these extremes are removed,
the RMSE lies within [17, 53] W m$^{-2}$, i.e. [1.7, 6.4] %, with a mean of 31 W m$^{-2}$ (3.4 %). Similarly to the bias
(3 W m$^{-2}$), the standard deviations of errors at the BSRN stations are very close to each other (17 and 19 W m$^{-2}$)
as well as the RMSE (17 and 19 W m$^{-2}$) (Table 11). One may note a tendency with some climate areas: the
smallest standard deviations are found at stations in climates BWh, BWk, Cfa, BSk, Cfb, and Csa, and range
between 12 and 23 W m$^{-2}$. Actually, the greater the mean $KT$, the smaller the standard deviation and the RMSE



(Tables 5 and 11). There is a slight trend with latitude which is likely related to the mean *KT* as the latter increases as the latitude decreases: the smaller the latitude, the smaller the standard deviation and the RMSE.

Standard deviations of errors vary very little throughout years at a given station (Figs. 7 and 8, left): relative changes are less than 0.5 % in 89 % of cases, and less than 1 % in all cases but 2. In addition, they are very close to each other at couples of close stations Pretoria (#30-31) and Durban (#36-37) for the same years. Similar observations are made for the RMSE as it is a quadratic combination of the standard deviations of errors and the bias. The widths of the intervals [13, 30] W m$^{-2}$ for the standard deviation throughout the stations and [1.4,

3.4] % for the relative standard deviation are 17 W m$^{-2}$ and 2.0 % respectively. This is very small: it can be reasonably assumed that the standard deviation of errors varies very little in space. Due to the influence of the bias, the widths of the RMSE intervals are larger: they are 26 W m$^{-2}$ and 4.7 % respectively. The relative standard deviation of errors is less than the expectations listed in Table 6 at all stations: except for the bias, the McClear outputs are compliant with the good quality standard of the World Meteorological Organization.


**Table 11. Standard deviations of errors (in W m$^{-2}$), RMSE (in W m$^{-2}$) and their values relative to the mean of the measurements (in %) for *G* at each station.**

| St. | Stand. dev. | Rel. stand. dev. | RMSE | Rel. RMSE | St. | Stand. dev. | Rel. stand. dev. | RMSE | Rel. RMSE | St. | Stand. dev. | Rel. stand. dev. | RMSE | Rel. RMSE |
|---|---|---|---|---|---|---|---|---|---|---|---|---|---|---|
| 1 | 24 | 2.7 | 24 | 2.7 | 16 | 28 | 3.4 | 44 | 5.3 | 31 | 22 | 2.5 | 35 | 3.9 |
| 2 | 21 | 2.4 | 21 | 2.4 | 17 | 26 | 3.1 | 44 | 5.1 | 32 | 26 | 3.0 | 63 | 7.1 |
| 3 | 27 | 3.1 | 27 | 3.1 | 18 | 25 | 2.9 | 42 | 4.8 | 33 | 17 | 1.7 | 22 | 2.2 |
| 4 | 21 | 2.4 | 29 | 3.3 | 19 | 25 | 2.9 | 37 | 4.4 | 34 | 18 | 2.0 | 24 | 2.7 |
| 5 | 24 | 2.7 | 37 | 4.1 | 20 | 26 | 3.0 | 41 | 4.7 | 35 | 18 | 1.7 | 19 | 1.8 |
| 6 | 25 | 2.9 | 31 | 3.6 | 21 | 27 | 3.3 | 53 | 6.4 | 36 | 22 | 2.4 | 23 | 2.6 |
| 7 | 19 | 2.1 | 28 | 3.1 | 22 | 24 | 2.8 | 46 | 5.4 | 37 | 18 | 2.0 | 18 | 2.0 |
| 8 | 16 | 1.7 | 21 | 2.3 | 23 | 24 | 2.8 | 43 | 5.1 | 38 | 18 | 1.8 | 19 | 1.8 |
| 9 | 25 | 2.7 | 25 | 2.7 | 24 | 31 | 3.7 | 50 | 5.9 | 39 | 14 | 1.4 | 20 | 2.1 |
| 10 | 20 | 2.0 | 21 | 2.1 | 25 | 22 | 2.6 | 39 | 4.5 | 40 | 13 | 1.3 | 22 | 2.1 |
| 11 | 22 | 2.4 | 25 | 2.7 | 26 | 22 | 2.2 | 23 | 2.3 | 41 | 13 | 1.4 | 16 | 1.7 |
| 12 | 21 | 2.2 | 28 | 3.0 | 27 | 23 | 2.6 | 33 | 3.7 | 42 | 20 | 2.1 | 23 | 2.4 |
| 13 | 21 | 2.4 | 23 | 2.6 | 28 | 17 | 1.7 | 17 | 1.7 | 43 | 23 | 2.4 | 27 | 2.8 |
| 14 | 27 | 3.2 | 40 | 4.6 | 29 | 22 | 2.4 | 43 | 4.7 | 44 | 15 | 1.5 | 16 | 1.7 |
| 15 | 30 | 3.2 | 48 | 5.2 | 30 | 20 | 2.1 | 24 | 2.6 | | | | | |





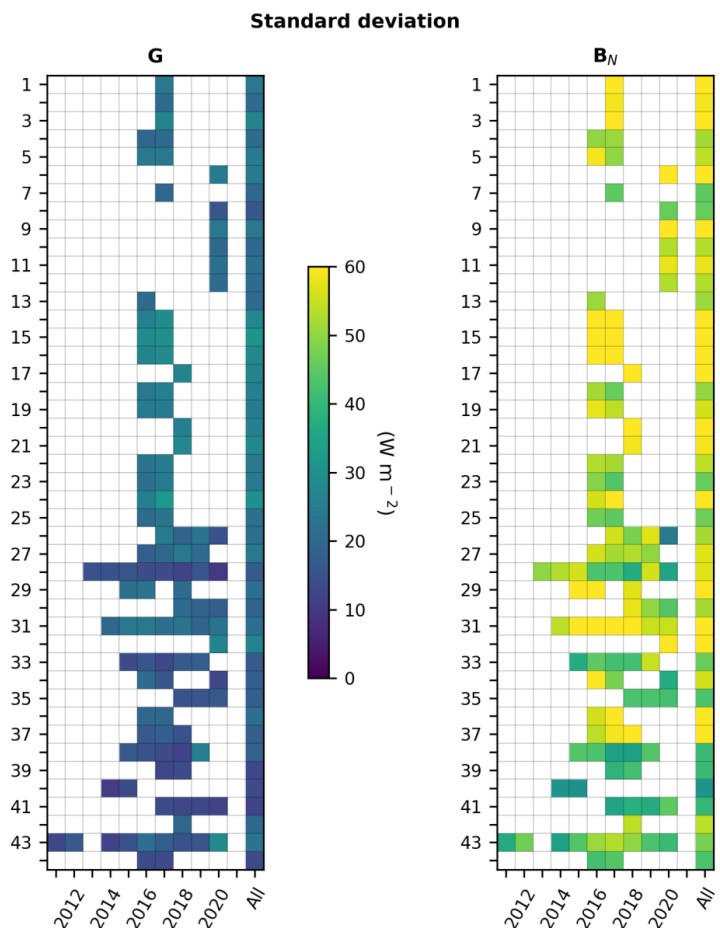

**Figure 7: Standard deviation of errors for *G* and *B_N* at each station for each year, and for the whole period (All). Numbers refer to the rank of the station in Table 1.**



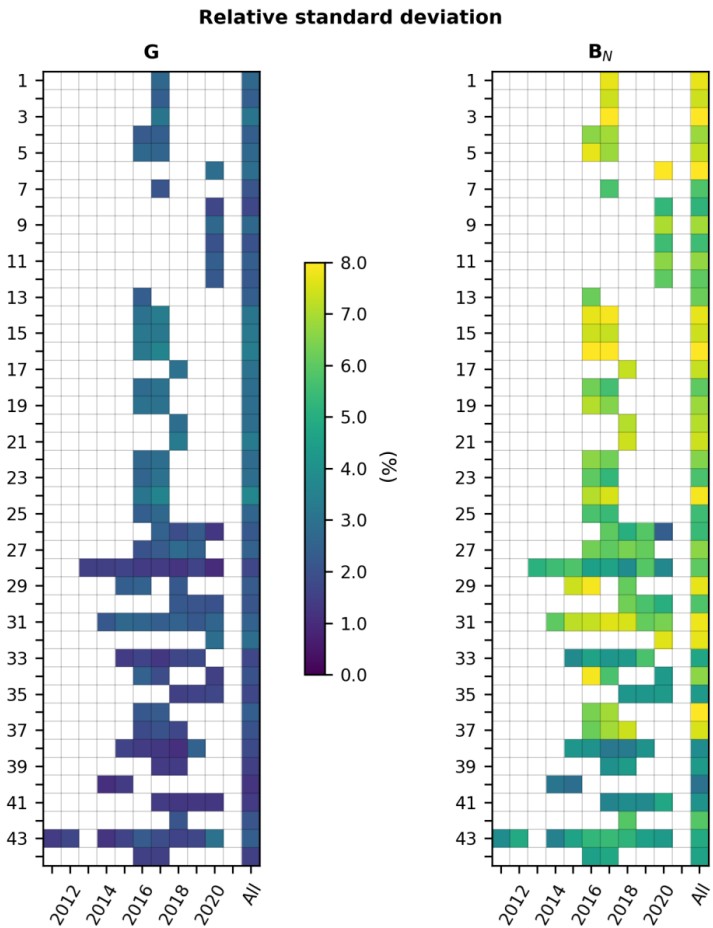

**Figure 8: Relative standard deviation of errors for *G* and *B_N* at each station for each year, and for the whole period (All). Numbers refer to the rank of the station in Table 1.**

### 4.6 Standard deviations of errors σ and RMSE for *B_N*

Table 12 gives the standard deviation of errors σ, the RMSE and their values relative to the mean of the measurements at each station for $B_N$. The rightmost graphs in Figures 7 and 8 exhibit respectively the standard deviation of errors and its relative value for $B_N$ at each station for each year and for the whole period. The standard deviation for $B_N$ ranges between 31 W m⁻² (3.0 %) at Graaff-Reinet (#40) and 70 W m⁻² at Gaborone (7.7 %, #29) and Pretoria-GIZ (7.9 %, #31) (Table 12). Once theses extremes removed, the standard deviation is within [40, 69] W m⁻², i.e. [4.1, 8.0] %, with a mean of 55 W m⁻² (6.4 %). The standard deviation is less than 60 W m⁻² (~7 % in relative value) at 31 stations out of 44 (70 % of the total) (Table 12). The RMSE for $B_N$ ranges between 32 W m⁻² (1.7 %) at Graaff-Reinet (#40) and 111 W m⁻² (13.2 %) at Witbank (#32) (Table 12). If these extremes are removed, the RMSE lies within [42, 88] W m⁻², i.e. [4.1, 11.0] %, with a mean of 63 W m⁻² (7.4 %). There is no trend with climate though one may note that the smallest standard deviations and RMSEs




are observed in climates BSk, BWk and Cfb. Though less pronounced than for $G$, the greater the mean $KT_{BN}$, the smaller the standard deviation and the RMSE (Tables 5 and 12). There is a slight trend with latitude which is likely related to the mean $KT_{BN}$ as the latter increases as the latitude decreases: the smaller the latitude, the smaller the standard deviation and the RMSE. Finally, the smaller the variance of the measurements, the smaller the standard deviation of errors and the RMSE.

Changes in standard deviations of errors from year to year at a given station are small in most cases as they are less than or equal to 10 W m$^{-2}$ in absolute value in 84 % of cases (Fig. 7, right). This ratio is 68 % for the RMSE (not shown). The standard deviations of errors respectively the RMSEs are very close to each other at couples of close stations Pretoria (#30-31) and Durban (#36-37) for the same years.

The widths of the intervals [40, 70] W m$^{-2}$ for the standard deviation throughout the stations and [3.9, 8.5] % for the relative standard deviation are 30 W m$^{-2}$ and 4.6 % respectively. They are moderate: it can be reasonably assumed that the standard deviation of errors varies fairly little in space. Due to the influence of the bias, the widths of the RMSE intervals are larger: they are 46 W m$^{-2}$ and 6.9 % respectively. The relative standard deviation of errors at any station is always above the expectations listed in Table 6: even excluding the bias, the McClear outputs for $B_N$ are not compliant with the good quality standard of the World Meteorological Organization.

**Table 12. Standard deviations of errors (in W m$^{-2}$), RMSE (in W m$^{-2}$) and their values relative to the mean of the measurements (in %) for $B_N$ at each station.**

| St. | Stand. dev. | Rel. stand. dev. | RMSE | Rel. RMSE | St. | Stand. dev. | Rel. stand. dev. | RMSE | Rel. RMSE | St. | Stand. dev. | Rel. stand. dev. | RMSE | Rel. RMSE |
|---|---|---|---|---|---|---|---|---|---|---|---|---|---|---|
| 1 | 61 | 7.7 | 69 | 8.7 | 16 | 66 | 8.5 | 70 | 9.0 | 31 | 70 | 7.9 | 72 | 8.0 |
| 2 | 59 | 7.5 | 70 | 8.8 | 17 | 60 | 7.3 | 60 | 7.3 | 32 | 64 | 7.6 | 111 | 13.2 |
| 3 | 68 | 8.6 | 72 | 9.1 | 18 | 50 | 6.0 | 73 | 8.8 | 33 | 46 | 4.7 | 51 | 5.2 |
| 4 | 52 | 6.9 | 54 | 7.1 | 19 | 56 | 6.9 | 58 | 7.1 | 34 | 55 | 6.6 | 60 | 7.1 |
| 5 | 55 | 7.3 | 69 | 9.3 | 20 | 60 | 7.1 | 60 | 7.1 | 35 | 43 | 4.3 | 44 | 4.4 |
| 6 | 66 | 8.1 | 68 | 8.3 | 21 | 59 | 7.4 | 65 | 8.1 | 36 | 69 | 8.0 | 70 | 8.0 |
| 7 | 45 | 5.7 | 53 | 6.8 | 22 | 53 | 6.5 | 79 | 9.7 | 37 | 64 | 7.5 | 65 | 7.6 |
| 8 | 46 | 5.3 | 47 | 5.4 | 23 | 47 | 5.7 | 72 | 8.9 | 38 | 41 | 3.9 | 43 | 4.1 |
| 9 | 62 | 7.0 | 78 | 8.8 | 24 | 63 | 7.8 | 88 | 11 | 39 | 42 | 4.3 | 43 | 4.5 |
| 10 | 53 | 5.5 | 57 | 6.0 | 25 | 46 | 5.6 | 74 | 8.8 | 40 | 31 | 3.0 | 32 | 3.1 |
| 11 | 58 | 6.6 | 58 | 6.7 | 26 | 53 | 5.5 | 56 | 5.8 | 41 | 40 | 4.1 | 42 | 4.4 |
| 12 | 53 | 6.0 | 53 | 6.0 | 27 | 56 | 6.6 | 66 | 7.7 | 42 | 54 | 5.9 | 56 | 6.1 |
| 13 | 51 | 6.2 | 64 | 7.7 | 28 | 57 | 6.0 | 80 | 8.4 | 43 | 48 | 4.9 | 51 | 5.2 |
| 14 | 62 | 7.8 | 73 | 9.1 | 29 | 70 | 7.7 | 76 | 8.3 | 44 | 43 | 4.7 | 46 | 5.0 |
| 15 | 64 | 7.4 | 82 | 9.5 | 30 | 52 | 5.9 | 53 | 5.9 | | | | | |

## 5 Possible explanations of the discrepancies

Discrepancies between ground-based measurements and McClear outputs are the results of the combination of uncertainties of several major sources, which are the uncertainties of the measurements already discussed in Section 2, the protocol of validation itself, the errors in the McClear model and of its inputs.

### 5.1 Protocol of validation

The protocol of validation used here is the same than those used in similar studies and is well-known. It presents two drawbacks because it compares quantities that are not exactly comparable in all aspects. McClear provides



total irradiance, i.e. integrated over the whole solar spectrum, namely [240, 4606] nm, while measurements by pyranometers are taken in a more limited range, often called broadband range, which is around [285, 2800] nm for pyranometers used in the BSRN, SAURAN and other networks. Simulations made with the radiative transfer model libRadtran show that the relative contribution of the irradiance in this interval [285, 2800] nm to the total irradiance is around 99 % and depends slightly on atmospheric properties. Hence by comparing measurements

acquired in BSRN-like interval and McClear estimates of total irradiance, one would expect an overestimation of approximately 1 %, i.e. of order of 5 W m$^{-2}$. We have assessed this point for $G$ and $B$ by comparing results published by Lefèvre et al. (2013) and Gschwind et al. (2019). Both works used the same dataset of measurements from 11 BSRN stations filtered for clear-sky instants. For the purpose of the comparison with BSRN measurements, Gschwind et al. computed two specific v1/v2 and v3 versions of McClear tailored to the BSRN spectral range [285, 2800] nm. Hence, the results of the BSRN-like v1/v2 in Tables 6 and 7 in Gschwind

et al. (Tables 8 and 9 for $B$) may be compared to those of the original v1 in Table 2 in Lefèvre et al. (Table 3 for $B$) to assess the influence of the spectral interval. As expected, the standard deviations of errors are similar as they range between 17 and 28 W m$^{-2}$ for the BSRN-like interval and between 18 and 27 W m$^{-2}$ for total irradiance for $G$ and between 32 and 43 W m$^{-2}$ for the BSRN-like interval and between 33 and 45 W m$^{-2}$ for total irradiance for $B$. The correlation coefficients are slightly greater for BSRN-like interval than for total irradiance

for both $G$ and $B$. Seven stations out of eleven exhibit a positive increase in bias for $G$ and four stations exhibit no change or a slight decrease of less than 4 W m$^{-2}$. A positive increase in bias from BSRN-like interval to total irradiance is observed at nine stations for $B$ and ranges from 4 to 6 W m$^{-2}$ and the two other stations show no change. The difference in relative bias is less than or close to 1 %. It can be concluded that part of the discrepancies between ground-based measurements and McClear outputs is due to the narrower spectral range of

the ground-based instruments, and that its magnitude depends on local conditions. Actual spectral interval of instruments should be taken into account for a more accurate validation of McClear and more generally of estimates of SSI.

The second drawback in the protocol concerns $B$ and $B_N$. Like most of the radiative transfer models, $B$ and $B_N$ are

modelled by libRadtran as if the sun were a point source. Therefore, they do not include the circumsolar radiation, i.e. the radiation coming from the vicinity of the direction of the sun, which is then entirely taken into account in the diffuse part $D$. On the contrary, pyrheliometers used in this study measure the radiation coming from the sun direction with a half-angle aperture of about 2.5° (Blanc et al., 2014a) and capture part of the circumsolar radiation whose magnitude is less than 10 W m$^{-2}$ in most clear-sky conditions (Oumbe et al., 2012)

and is fairly similar to the uncertainty of the instruments though the magnitude may be greater depending on the type of aerosol and its optical depth (Blanc et al., 2014a; Eissa et al., 2018; Gueymard, 2010; Oumbe et al., 2013). This is also true for the other instruments used in estimating $B_N$ (Table 3). As no correction to McClear is brought for the contribution due to the circumsolar area, one may expect McClear to underestimate $B$ and $B_N$. The variance should also be underestimated. These underestimations of mean and variance of $B_N$ do not clearly

appear in this study as they are observed at 17 stations only out of 44 for the mean and 20 stations only for the variance. It is obvious that other uncertainties are more important. In addition, the correlation coefficient is affected because the variability of the circumsolar radiation is not taken into account in the direct component in McClear. As a consequence, one may expect very low correlation coefficient though it depends on the local atmospheric conditions.





### 5.2 Uncertainties of the inputs to McClear and of the McClear model itself

Discrepancies between ground-based measurements and McClear outputs are due to the combination of the uncertainties of the measurements themselves, those of the input to the McClear model and those of the model itself. Aerosol loading and type, water vapor amount and atmospheric profiles of atmospheric constituents have a

great influence on the SSI in clear sky conditions. In this respect, the McClear v3 model has several shortcomings underlined by Gschwind et al. (2019). One of them is the use of prescribed vertical profiles of temperature, pressure, density, and volume mixing ratio for gases as a function of altitude taken from the AGFL (USA Air Force Geophysics Laboratory): tropics (afglt), mid-latitude summer and winter (afglmls and afglmlw), and sub-Arctic summer and winter (afglss and afglsw), as implemented in libRadtran and not actual ones. Using

such prescribed vertical profiles instead of actual ones may lead to differences of a few percent in irradiance at the surface as shown in numerical simulations (Oumbe et al., 2008). As a whole, the McClear model allocates the sub-Arctic profiles to polar and cold climate zones EF, ET, Df, Ds, and Dw, the mid-latitude profiles to arid and temperate climates BS, BW, Cf, Cs and Cw, and the tropical profile to tropical climates Af, Am and Aw (Lefèvre et al., 2013). The map of climates used in McClear is very coarse though care has been taken to avoid

spatial discontinuity (Gschwind et al., 2019) and stations may be allotted to a wrong climate type. Because the prescribed profiles are typical profiles, they may differ from the actual ones. A systematic mismatch between the prescribed vertical profile and the actual ones may partly explain the link between biases and variances, and climates and high elevation.

McClear uses the description of the aerosol properties taken from the database OPAC in libRadtran, namely sea

salt, dust, organic matter, black carbon and sulfates aerosol species. This description may be too coarse to precisely describe the aerosol properties and their influence on the SSI. The mapping between CAMS and OPAC species adopted in McClear may also account for a part of the error. McClear also uses prescribed vertical profiles of aerosol loads instead of actual ones and this may lead to differences of a few percent in irradiance at the surface (Fountoulakis et al., 2022).

The parameters *fiso*, *fvol*, and *fgeo* (Schaaf et al., 2002) that describe the bidirectional reflectance distribution functions (BRDF) of the ground are several years-averaged values for each month taken from Blanc et al. (2014b) and the actual reflection properties are not taken into account. The reflection properties and their spectral variation have an important influence on the diffuse part of the SSI (Oumbe et al., 2008) and exhibit very high spatial variation on a pixel basis and daily to seasonal temporal variations, thus adding to the

discrepancies between McClear outputs and measurements. Differences between the elevation of the CAMS cell and the actual elevation of the station are taken into account by a linear interpolation in clearness index (Lefèvre et al., 2013).

Also to be considered is the solar irradiance impinging at the top of atmosphere at normal incidence $E0_N$. It varies with the changing position of the Earth on its orbit and this is taken into account in McClear in an accurate

way via the SG2 algorithm (Blanc, Wald, 2012). But the activity of the sun itself includes changes in the intensity of the emitted solar radiation. The solar activity exhibits a nearly periodic 11-year cycle, each cycle being characterized by the number and size of sunspots, flares, and other manifestations. The solar cycle has a limited influence on $E0_N$, of order of 0.1 %. Said differently, average changes during a cycle are small and of



order of 1 W m$^{-2}$. However, day-to-day changes in $E0_N$ are greater and may reach 5 W m$^{-2}$, i.e. approximately

0.4% of the total solar irradiance $E_{TSI}$ (Kopp and Lean, 2011). For given atmosphere and ground properties, the greater $E0_N$, the greater $B_N$, $B$, $D$ and $G$. Though small, these daily changes in $E0_N$ add to the discrepancies between McClear and the measurements as the latter take these daily changes into account while McClear does not.

Inputs from CAMS, namely the total column contents in ozone and water vapor, the total aerosol optical depth

and the partial optical depth for sea salt, dust, organic matter, black carbon and sulfates aerosol species, all of them at 550 nm, are given for cells of several km in size and once a day or every 3 h. They are resampled to the selected location by spatial bilinear interpolation and resampled in time to the desired summarization. The results do not contain the sub-cell and 1-min variabilities of $G$ and $B_N$. Thus the exact atmospheric effects on the incident solar radiation over a specific site cannot be captured and this adds to the discrepancies between the

McClear outputs and the measurements. The magnitude of the discrepancy cannot be predicted as it depends on the atmospheric conditions experienced by each station every 1 min. As an example, Zieger et al. (2010) report noticeable changes in single scattering albedo with relative humidity for several OPAC species. If relative humidity is assumed too large, then the single scattering albedo is overestimated, yielding an underestimation in the diffuse component of the SSI and thus in $G$ and in the circumsolar portion in the measurements of $B_N$. Such

changes in relative humidity may occur at short space and timescales and cannot be accounted for in CAMS outputs. In another example of the variability within a cell of a few km in size, Wald and Baleynaud (1999) attribute noticeable changes in atmospheric transmittance detected in high-resolution satellite imagery to changes in PM loads due to vehicle traffic in cities at scale of 100 m, thus affecting $G$ and $B_N$. The variation of the influencing variables within the cell affect the statistical quantities in the comparison.For example, Oumbe et al.

(2012) report changes in the standard deviations of errors in $B_N$ up to 18 % over distances less than 100 km due to the variability of aerosol loads in the United Arab Emirates. In another example comparing 1 min measurements of $G$ and $B_N$ made in clear-skies conditions at 9 stations in Singapore in a small area of 6x3 km² against model-derived estimates, Sun et al. (2022) found that the relative RMSE may vary up to a factor of 3 for $G$ and $B_N$ between the stations. Similar results are reported in Perez et al. (1997) and Zelenka et al. (1999) with

hourly means of $G$ measured at sites less than 50 km apart. In another example, the comparison made by Qin et al. (2022) between satellite-derived SSI and 10-min ground measurements suggests that a large part of apparent validation errors may be due to the mismatch in spatial sampling, often exceeding 50 % in the case of $B_N$. In a comparison of 1 h and 10 min measurements of $G$ and $B_N$ respectively, Eissa et al. (2015a) and Lefèvre and Wald (2016) report that the inter-stations correlation is greater in McClear estimates than in measurements for

both $G$ and $B_N$ and advocate that this may be attributed to the stronger than actual correlation in space and time of inputs from CAMS due to their coarse spatial and temporal resolutions.  These examples stress the large spatial variability of the SSI and the influence of the coarse resolution in time and space of several inputs to McClear.

We have explored the influences of several variables by means of boxplots of the differences between the

McClear estimates and the measurements. As examples, Figs 9 and 10 exhibit boxplots for $G$ and $B_N$ respectively at the station Gobabeb (#28) for different classes of $\theta_S$, different classes of ground albedo read from McClear outputs and different classes of readings from CAMS, namely total column contents in ozone and water vapor and optical depth of aerosols at 550 nm. We selected this station because it is part of the BSRN reference





network BSRN. Similar boxplots were drawn for $G$, $B_N$, $KT$, and $KT_{BN}$ at each station to assess the influence of

these quantities. All graphs at each station are available as supplementary material.

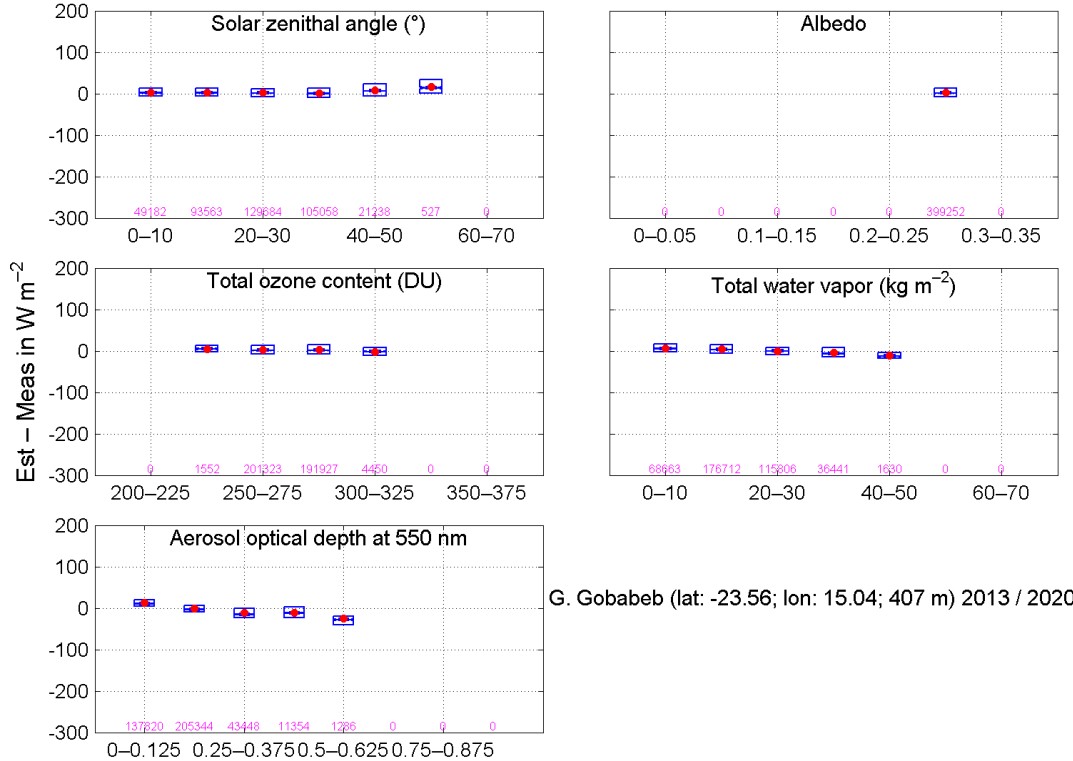

**Figure 9: Boxplots of the differences between the McClear estimates and the measurements of *G* at the BSRN station Gobabeb (#28, climate BWh) for several classes of various variables. In each boxplot, the red point is the mean while**
**the lower, middle and upper lines are respectively the 25, 50 and 75 percentiles. The number of samples is given for each class. Only classes with at least 50 samples are drawn.**





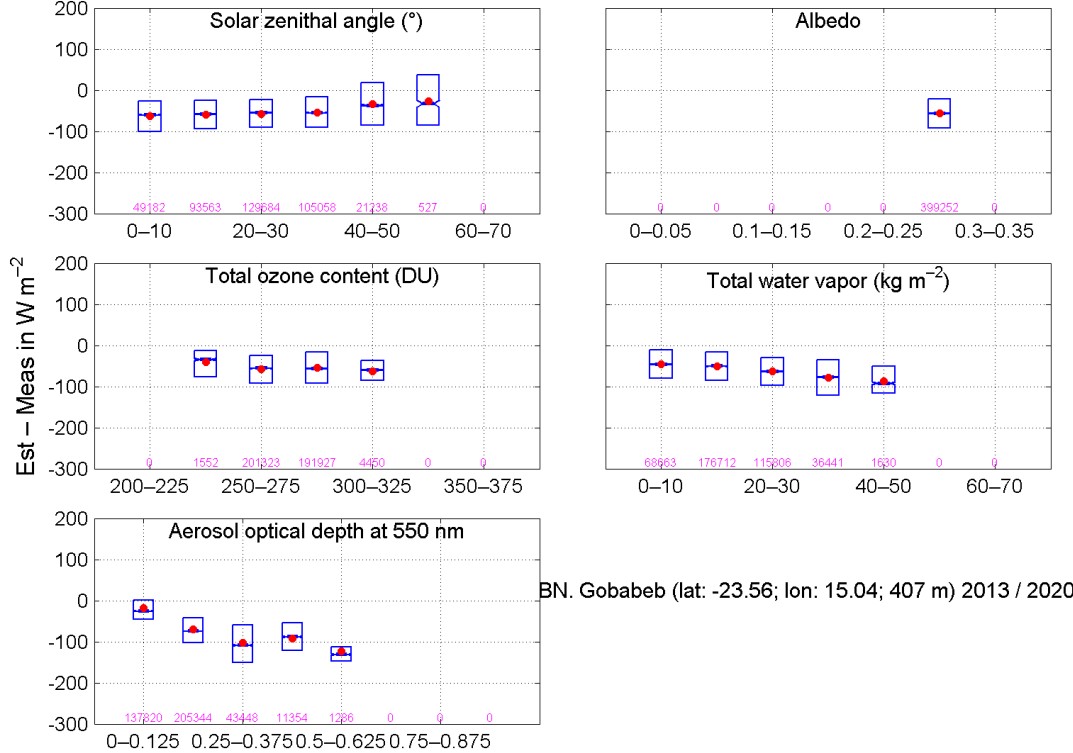

**Figure 10: Boxplots of the differences between the McClear estimates and the measurements of $B_N$ at the BSRN station Gobabeb (#28, climate BWh) for several classes of various variables. In each boxplot, the red point is the mean while the lower, middle and upper lines are respectively the 25, 50 and 75 percentiles. The number of samples is given for each class. Only classes with at least 50 samples are drawn.**

The solar zenithal angle $\theta_S$ is computed with great accuracy using the SG2 algorithm and exhibit a very small uncertainty. In addition, a great deal of efforts was made to accurately model the influence of $\theta_S$ in McClear v3 and graphs in Gschwind et al. (2019) show that the error is very small whatever $\theta_S$. Hence, any error observed in boxplots for $\theta_S$ originate from errors in other inputs, namely total column contents in ozone and water vapor, aerosol optical depth at 500 nm and descriptors of the reflection of the ground as well as uncertainties in the McClear model. The ground albedo is a side-product of McClear. It depends on the BRDF as well as on $G$ and $B_N$ (Lefèvre et al., 2013) and includes errors on the BRDF parameters and model as well as other errors on other inputs to McClear. As such, the ground albedo is not an input to McClear and is an ouput of the McClear service. Nevertheless, similarly to the solar zenithal angle $\theta_S$, boxplots of the differences between the McClear estimates and the measurements for different classes of ground albedo have been included in Figs 9 and 10.

As a whole, the boxes are very narrow to narrow for $G$ (e.g. Fig. 9) which means that the intra-class variances of errors are small contrary to $B_N$ for which boxes are much wider (e.g. Fig. 10). This is in line with the standard deviations of errors for $G$ and $B_N$ discussed in the previous section.

The case of Gobabeb does not represent the other stations. Patterns of changes of errors with the variables depend strongly on the stations though several rules may be found. As a whole, changes of errors with total





column content in ozone or water vapor for $G$ are small at each station (see e.g. Fig. 9). Changes of errors are

more pronounced for $B_N$ (see e.g. Fig. 10) but there is no clear pattern as the changes depend upon the stations. Patterns of changes of errors with aerosol optical depth (AOD) exhibit some consistency with climates which is not the case for the total column contents in ozone and water vapor. As a whole, the errors for $B_N$ are positive or close to 0 at low AOD and decreases as the AOD increases, becoming more and more negative (see e.g. Fig. 10). The decrease may be small (temperate climate Cwb), moderate like in Fig. 10 (tropical climates Am and Aw,

desert climates BWh and BWk, and temperate climate Cwa) or strong (steppe climates BSh and BSk, and temperate climates Cfa, Cfb, and Csa). The patterns are more pronounced for $G$ for each climate. Stations experiencing climate Am exhibit negative errors and a slight decrease (more and more negative) as the AOD increases, in line with the underestimation of means. The errors are positive and fairly independent of the AOD in climate Aw in line with the overestimation of means. Errors are close to 0 and decrease slightly in climates

BSh and BSk, while they are close to 0 and independent of the AOD in climates BWh (Fig. 9) and BWk and gently decreasing in climates Cfa, Cfb and Csa; these patterns are consistent with the correct estimation of the means. In climates Cwa and Cwb, the errors are close to 0 or slightly positive and constant but slightly decrease when the AOD is greater than 0.5-0.625, in line with the overestimation of the means.

The strong dependence with AOD observed at many stations in the Supplement is quite understandable given the

1-min resolution of the measurements. The greatest CAMS AOD are likely quite rare cases of strongest episodes that cannot be represented by point-wise 1-min measurements. One may also consider that in this kind of comparison, it is typical to see that smallest values are overestimated and greatest underestimated. As discussed earlier, part of the such bias should be present even if both McClear and ground-based measurements were fully accurate since they do not have the same temporal and spatial scales.

Table 13 summarizes the patterns of errors with a AOD for each climate and includes the errors on means and variances for both $G$ and $B_N$. Details are given in the supplementary material. This table demonstrates clear links between climates and the errors in means, variances and for the AOD classes. We believe that there is no direct link between the errors on the SSI and the climates as the latter are defined from long time series of precipitation and temperature measured at stations (Peel et al., 2007). Links between climates and errors are indirect and result

from combinations of shortcomings in the McClear model and gross errors in aerosol properties modelled as input to McClear, including interactions between hydrophilic species and relative humidity at each altitude and differences between the standard atmospheres used and the actual ones.

**Table 13. Summary of errors on means and variances and for AOD classes for each climate.**

| Climate | Mean | Variance | Errors for AOD classes |
|---|---|---|---|
| Am | $G$: underestimation | $G$: underestimation | $G$: <0, slight decrease and more and more <0 |
| | $B_N$: strong overestimation | $B_N$: - | $B_N$: >0 at low AOD, decrease and more and more <0 |
| Aw | $G$: overestimation | $G$: underestimation | $G$: >0, constant |
| | $B_N$: - | $B_N$: - | $B_N$: >0 at low AOD, decrease and more and more <0 |
| BSh, BSk | $G$: correct | $G$: underestimation | $G$: ~0 at low AOD, slight decrease, more and more <0 |
| | $B_N$: - | $B_N$: - | $B_N$: ~0 at low AOD, strong decrease, more and more <0 |
| BWh, BWk | $G$: correct | $G$: correct | $G$: ~0 and constant |
| | $B_N$: - | $B_N$: - | $B_N$: >0 at low AOD, decrease and more and more <0 |
| Cfa, Cfb, Csa | $G$: correct | $G$: correct | $G$: close to 0 though often >0, slight decrease, more and more <0 |
| | $B_N$: - | $B_N$: - | $B_N$: >0 at low AOD, strong decrease and more and more <0 |





| | | | |
|---|---|---|---|
| Cwa | $G$: strong overestimation | $G$: correct | $G$: fairly constant and >0, <0 at AOD>0.625 |
| | $B_N$: strong overestimation | $B_N$: - | $B_N$: most often >0, decrease |
| Cwb | $G$: strong overestimation | $G$: underestimation | $G$: ~0, fairly constant though slight decrease at AOD>0.5 |
| | $B_N$: - | $B_N$: - | $B_N$: slight decrease |


Oumbe et al. (2012, 2013) and Eissa et al. (2015b) compared the AODs at 550 nm read from the CAMS reanalysis to AERONET measurements in the United Arab Emirates (climate BWh). They found that CAMS overestimates the AOD and that the bias increases as the AOD increases. The relative bias in AOD is in the range 10-20 %. These authors wrote that errors in AOD are the major source of errors in $G$ and $B_N$. The bias in $G$

is negative and equal to −3 % in relative value with a relative RMSE of 5 % (Eissa et al., 2015b; Oumbe et al., 2012). The underestimation in $B_N$ is more pronounced and is more and more negative as the AOD increases (Oumbe et al., 2012), in agreement with our findings. Oumbe et al. (2015) underline the high variability of the bias in AOD between the stations.

Besides these above-cited early works offering detailed analyses on the United Arab Emirates and BWh climate,
several articles, reports and working papers have been published in the past few years on the comparison of CAMS AOD and AERONET measurements. Several important activities take place within the AeroCom project (https://aerocom.met.no, last access: 2022-08-11) which assembles a large number of observations and results from many global models to document and compare state-of-the-art modeling of the aerosols on a global scale. The website https://aerocom-classic.met.no/cams-aerocom-evaluation/ (last access: 2022-08-11) provides
interactive graphs comparing the CAMS total AOD and its components to measurements at several stations worldwide. Unfortunately, the AERONET stations available in this website do not match ours, except Hanimaadhoo (#5, climate Am) and Pretoria CSIR (#30, climate Cwb). At Hanimaadhoo, CAMS estimates are close to AERONET measurements, and one may expect small biases in $G$ and $B_N$, which is not the case as Tables 7 and 8 report an underestimation of −20 W m$^{-2}$ for $G$ and an overestimation of 14 W m$^{-2}$ for $B_N$.
Measurements at Pretoria cover an earlier period than ours during which a strong overestimation of measurements by CAMS is observed. Particulate organic matter is the greatest contributor to the CAMS AOD. If one assumes that this overestimation stands also for the period 2018-2020, one would expect an underestimation of both $G$ and $B_N$ but Tables 7 and 8 indicate a slight overestimation of both. The CAMS website https://global-evaluation.atmosphere.copernicus.eu/aerosol/aod-aeronet (last access: 2022-08-27) provides similar interactive
graphs comparing the CAMS total AOD to measurements at several stations worldwide. Though this site is dedicated to the validation of the forecast 1 day ahead, it provides an insight about the accuracy of CAMS AOD. There are three stations matching ours: Hanimaadhoo (#5, climate Am), Gobabeb (#28, BWh) and Durban-KZW (#36, Cfa) but the periods are 20020 only or 2020-2021. Comparisons with our results on SSI may be done if one assumes that the results on AOD forecasts stand for the previous periods. At Hanimaadhoo, CAMS forecasts are
close to AERONET measurements, and one may expect small biases in $G$ and $B_N$, which is not the case as discussed earlier about AEROCOM results. Overestimations of measurements by CAMS are observed at Gobabeb and Durban. One would expect underestimations of both $G$ and $B_N$. Table 7 shows small positive biases for $G$ and Table 8 reports negative biases for $B_N$.

Of particular interest here, are the articles of Gueymard and Yang (2020) and Salamalikis et a. (2021). Both
report that the AERONET AODs tend to be underestimated by CAMS in regions dominated by coarse aerosols from mineral dust and biomass burning, including our area of study. The underestimation is usually small but is highly variable in space and time. Gueymard and Yang (2020) performed a classification of the errors in AOD at



550 nm (bias and RMSE) as a function of the climate and evidenced a link between the errors and the climate though they reported a high variability of errors within each climate (see their Figs 6 and 9). Though there are some limitations to the direct link between errors in estimating AOD and errors in estimating $G$, we observe that their results may partly explain ours. In climate Am, the AOD tends to be overestimated by CAMS (see their Table 4) which is in agreement with an underestimation of $G$ as reported in Table 13. On the contrary, the underestimation of AOD in climates Aw or Cwa (their Table 4) corresponds to an overestimation in $G$ (Table 13). However, the relationship is not so strong. As a whole, the AOD is more or less overestimated at the AERONET stations in climates BSh, BSk, BWh, BWk, Cfa, Cfb, and Csa, while at our stations in the same climates, the estimation of $G$ is correct (Table 13).

**6 Comparison between previous published works and ours**

In this section, we have assembled the performances of McClear reported from similar previous works to assess whether our findings are in agreement with similar published works regarding the range of values for each indicator and the variability of these indicators between sites. It is also an opportunity to gather our findings and other works to establish a general overview of the performances of McClear.

Several works have been published comparing McClear outputs and in situ measurements for various summarizations. As the focus of this paper is on the assessment of performances at individual sites, we have retained in the following only those works allowing such a comparison. Results from several works cannot be compared to ours because:

- the period is very limited such as Dev et al. (2017) or unsuitable for comparison like the official CAMS validation reports (Lefèvre, 2021) which deal with trimesters only,
- the work uses a special version of McClear tailored to pyranometer spectral range (Gschwind et al., 2019),
- the work focuses on solar forecasting in all skies conditions and does not assess the performances of McClear per se (Yang, 2020),
- the lack of quantitative measures of performance at each station (Chen et al., 2020; Sun et al., 2019, 2021).

Table 14 lists the published works comparing McClear outputs and in situ measurements for summarization 1 min, 10 min, and 1 h. A letter code was allotted to each work to ease the reading of the following tables.

**Table 14. List of selected published works comparing McClear outputs and in situ measurements for summarization 1 min, 10 min and 1 h (column time) and some of their characteristics. The letter code is used in the following tables.**

| Time | Letter code | Authors | Area | Period | Version | Number of stations | Quantities |
|---|---|---|---|---|---|---|---|
| 1 min | (a) | Lefèvre et al. (2013) | World | 2005-2008 | v2 | 11 | $G, B$ |
| | (b) | Cros et al. (2013) | La Réunion Island | 2010-2012 | v2 | 1 | $G$ |
| | (c) | Ceamanos et al. (2014) | Europe, Middle East, North Africa | 2011 | v2 | 7 | $G, B$ |
| | (d) | Zhong and Kleissl (2015) | California, Nevada, USA | 2009-2011 | v2 | 5 | $G, B_N$ |
| | (e) | Alani et al. (2019) | Benguerir (Morocco) | 2015-2017 | v2 | 1 | $G, B_N$ |



| | | | | | | |
|---|---|---|---|---|---|---|
| | (f) | Antonanzas-Torres et al. (2019) | Europe | 2014 | v3 | 2 | $G, B_N$ |
| | (g) | Tahir et al. (2022) | Pakistan | 2015-2016 | v3 | 9 | $G$ |
| 10 min | (o) | Eissa et al. (2015b) | United Arab Emirates (UAE) | 2012 | v2 | 7 | $G, B_N$ |
| | (p) | Lefèvre and Wald (2016) | Israel | 2006-2011 | v2 | 3 | $G, B, B_N$ |
| 1 h | (t) | Cros et al. (2013) | Corsica Island | 2010-2011 | v2 | 1 | $G$ |
| | (u) | Eissa et al. (2015a) | Egypt | 2004-2009 | v2 | 7 | $G, B_N$ (only 2 sites) |
| | (v) | Ineichen (2016) | Europe, Africa | 2004-2013 | v2 | 22 | $G, B_N$ |
| | (w) | Salamalikis et al. (2022) | Greece | 2014-2020 | v3 | 1 | $G$ |

**6.1 Limitations in the comparison**

Comparing similar published performances with the present work is not so easy for various reasons and some of them create limitations to the comparisons between our findings and these works and also among these works. As discussed in section 2.3, the three versions v1, v2 or v3 offer similar results and consequently the use of different versions in the selected works is not a noticeable limitation.

The use of different time periods in the compared works is a first limitation. Using sets of measurements spanning several years, Lefèvre and Wald (2016), Lefèvre et al. (2013), or Gschwind et al. (2019) observed that the statistical indicators of McClear slightly vary from one year to another for both $G$ and $B$ or $B_N$, though no clear trend may be noticed. Changes in bias or standard deviations of errors of order of 10 W m$^{-2}$ for $G$ or $B_N$ are reported by these authors while changes in correlation coefficients are very small. These results are mostly in 895 line with our findings in section 4. Such changes with time period must be kept in mind when comparing results obtained for different years at the same station.

Another limitation is that some authors performed validation of $B$ or $D$ whereas others dealt with $B_N$. As the relationship between $B$ and $B_N$ is not linear because of the presence of $\cos(\theta_S)$, comparing the different results reveals difficult. For example, the correlation coefficients for $B$ are usually very large whereas those for $B_N$ are 900 smaller due to the small variances of the measurements and estimates of $B_N$ in clear-sky conditions.

The different selection of stations is a further limitation. These differences can be appreciated in the tables in the following section where we have reported all results found at individual stations by the various authors and are discussed later.

Another limitation originates from different algorithms for the selection of clear-sky instants adopted by the 905 various authors. Table 15 lists the algorithms adopted in the works used here. To obtain a first assessment of the influence of this choice, we have compared the McClear performances found by Ceamanos et al. (2014, their Tables 6 and 4) and Lefèvre et al. (2013, their Tables 2 and 3) at the same three BSRN stations: Carpentras, Sede Boqer and Tamanrasset. To select clear-sky instants in the measurements, Ceamanos et al. (2014) used the cloud mask provided by the NWC-SAF (http://www.nwcsaf.org/, last access: 2022-08-12), which is built and released 910 every 15 min from infrared observations acquired aboard Meteosat satellites. This strategy provides more clear-



sky instants than the algorithm of Lefèvre et al. (2013) which is more selective. Correlation coefficients are similar between both works. The difference in bias between both works ranges between 3 and 7 W m$^{-2}$ for $G$ and between 0 and 10 W m$^{-2}$ for $B$. That in standard deviation of errors ranges between 4 and 10 W m$^{-2}$ for $G$ and between 3 and 35 W m$^{-2}$ for $B$ while that in RMSE ranges between 3 and 10 W m$^{-2}$ for $G$ and between 5 and 35

W m$^{-2}$ for $B$. Such differences may be large depending on the case. This result underlines the local influence of algorithm for the selection of clear-sky instants and should be born in mind when comparing published works.

**Table 15. List of algorithms used for detecting clear-sky instants.**

| Authors | Algorithm |
|---|---|
| (a) Lefèvre et al. (2013) | Own algorithm |
| (b, t) Cros et al. (2013) (1 min, 1 h) | Own algorithm inspired from Lefèvre et al. (2013) and Ineichen (2006) |
| (c) Ceamanos et al. (2014) | Cloud mask estimated from infrared observations acquired aboard Meteosat satellites |
| (d) Zhong and Kleissl (2015) | Algorithm of Long and Ackerman (2000) |
| (e) Alani et al. (2019) | Algorithm of Lefèvre et al. (2013) |
| (f) Antonanzas-Torres et al. (2019) | Visual inspection of data |
| (g) Tahir et al. (2022) | Algorithm of Lefèvre et al. (2013) |
| Mabasa et al. (2021) | Use of the ERA5 reanalysis in a first step and then, a combination of a modified version of the algorithm of Reno and Hansen (2016) and McClear outputs |
| (o) Eissa et al. (2015b) (10 min) | Modified version of the algorithm of Long and Ackerman (2000) |
| (p) Lefèvre and Wald (2016) (10 min) | Algorithm of Lefèvre et al. (2013) |
| (u) Eissa et al. (2015a) (1 h) | Modified version of the algorithm of Lefèvre et al. (2013) |
| (v) Ineichen (2016) (1 h) | Own algorithm |
| (w) Salamalikis et al. (2022) (1 h) | Own algorithm |

Another limitation is the summarization of the measurements, which is 1 min, or 10 min or 1 h (Table 14). Tables 16 and 17 report statistical indicators obtained at the same station for different summarizations for $G$ and $B$ or $B_N$. This comparison has itself some limitations as the selection of clear-sky instants may be different, it is limited to a small number of sites and periods of comparison are different. Having these in mind, one observes that for $G$, the bias has the same sign between summarizations and is often greater at 1 h than at 1 min (Table

16). The differences between 1 min and 1 h are close to those between different works at the same summarization at the same site and close to those mentioned earlier regarding different selection of clear-sky instants or different periods. The standard deviation at 1 h is less than that at 1 min which is expected as it reflects the lower variability of the SSI at 1 h compared to 1 min. The RMSEs are close between summarizations. There is only one station for $B$ or $B_N$ to compare summarizations (Table 17). The correlation

coefficient is less at 10 min than at 1 min which was expected because the influence of $\cos(\theta_S)$ on the correlation coefficient for $B$ is less in the former case. From their own experience in validation of SSI against ground-based measurements, the present authors have observed that as a whole, the correlation coefficient for $G$ and $B$ decreases as the summarization increases from 1 min to 1 day. The bias (underestimation) is more pronounced at 10 min than at 1 min. As for $B_N$, the statistical indicators are similar between 1 min and 1 h. The differences

between statistical indicators observed at different summarizations are close to those observed between the various authors for the same summarization at the same site. Consequently, results from different summarizations may be compared with limitations similar to those reported about the algorithm for selecting clear-sky instants or different periods. However, for the sake of clarity, we have presented the statistical indicators found in literature for each summarization in the following section.





**Table 16. Comparison of results obtained for *G* at different summarizations at the same station.**

| Station | Authors | Mean of measurements (W m⁻²) | Correl. coeff. | Bias (W m⁻²) | Standard deviation (W m⁻²) | Rel. bias (%) | Rel. standard deviation (%) | RMSE (W m⁻²) | Rel. RMSE (%) |
|---|---|---|---|---|---|---|---|---|---|
| Toravere | (c) 1 min | 484 | 0.995 | 2 | 18 | 0.4 | 3.7 | 19 | 3.9 |
|  | (v) 1 h | 500 | N/A | 12 | 14 | 2.4 | 2.8 | 18 | 3.6 |
| Cabauw | (c) 1 min | 446 | 0.995 | 8 | 21 | 1.8 | 4.7 | 23 | 5.2 |
|  | (h) 1 min | - | 0.995 | - | - | 1.3 | - | - | 3 |
|  | (v) 1 h | 543 | N/A | 13 | 17 | 2.4 | 3.1 | 21 | 3.9 |
| Payerne | (a) 1 min | 596 | 0.995 | 22 | 22 | 3.7 | 3.7 | 29 | 4.9 |
|  | (v) 1 h | 604 | N/A | 22 | 17 | 3.6 | 2.8 | 28 | 4.6 |
| Carpentras | (a) 1 min | 629 | 0.995 | 20 | 21 | 3.2 | 3.3 | 31 | 4.9 |
|  | (c) 1 min | 553 | 0.995 | 13 | 17 | 2.4 | 3.1 | 22 | 4.0 |
|  | (h) 1 min | - | 0.998 | - | - | 2.0 | - | - | 3 |
|  | (v) 1 h | 587 | N/A | 14 | 18 | 2.4 | 3.1 | 23 | 3.9 |
| Sede Boqer | (a) 1 min | 785 | 0.995 | 12 | 24 | 1.5 | 3.1 | 27 | 3.4 |
|  | (c) 1 min | 636 | 0.995 | 9 | 28 | 1.4 | 4.4 | 30 | 4.7 |
|  | (p) 10 min | 838 | 0.998 | 2 | 30 | 0.2 | 3.6 | 30 | 3.6 |
|  | (v) 1 h | 744 | N/A | 11 | 24 | 1.5 | 3.2 | 26 | 3.5 |
| Tamanrasset | (a) 1 min | 791 | 0.990 | 8 | 18 | 1.0 | 2.3 | 20 | 2.5 |
|  | (c) 1 min | 650 | 0.995 | 5 | 28 | 0.8 | 4.3 | 28 | 4.3 |
|  | (v) 1 h | 672 | N/A | 16 | 17 | 2.4 | 2.5 | 23 | 3.4 |

**Table 17. Comparison of results obtained for *B* or $B_N$ at different summarizations at the same station.**

| Station | Authors | Mean of measurements (W m⁻²) | Correl. coeff. | Bias (W m⁻²) | Standard deviation (W m⁻²) | Rel. bias (%) | Rel. standard deviation (%) | RMSE (W m⁻²) | Rel. RMSE (%) |
|---|---|---|---|---|---|---|---|---|---|
| Sede Boqer (*B*) | (a) 1 min | 667 | 0.975 | −48 | 39 | −7.2 | 5.8 | 62 | 9.3 |
|  | (c) 1 min | 527 | 0.985 | −51 | 51 | −9.7 | 9.7 | 72 | 13.7 |
|  | (p) 10 min | 724 | 0.968 | −66 | 45 | −9.1 | 6.2 | 80 | 11.0 |
| Sede Boqer ($B_N$) | (p) 10 min | 878 | 0.781 | −68 | 48 | −7.7 | 5.5 | 83 | 9.5 |
|  | (v) 1 h | 868 | N/A | −72 | 50 | −8.3 | 5.8 | 88 | 10.1 |

## 6.2 Comparison with Mabasa et al. (2021)

First of all, we have compared our results to those obtained by Mabasa et al. (2021) at 13 stations measuring *G* in South Africa every 1 min, from 2013 to 2019. Table 18 reports the results from these authors which are given by large intervals and may render the comparison unprecise. Only two stations in Mabasa et al. (2021) are in our list: Durban (#36-37) and De Aar (#38). Their correlation coefficients are greater than 0.995 and greater than ours: 0.978 to 0.991. Regarding relative biases and RMSEs, ours are similar to theirs at De Aar. At Durban, our relative biases are 1.0 % and −0.2 % versus [2, 5] %, for theirs, and our relative RMSEs are 2.4 % and 2.0 % versus [5, 10] %) for theirs. We have also compared their results at their 13 stations (Table 18) to ours at our 16 southernmost stations, from #27 to #44, excluding Witbank (#32) since the two sets of stations are located in the same region. Our correlation coefficients are comprised between 0.966 and 0.991 and are a bit less than theirs which are greater than 0.995. Except at three stations, their relative biases are positive and, in the intervals, [0, 2] % at two stations and [2, 5] % at 8 stations. Except at three stations (#35, 37, 44), our relatives biases are positive and, in the interval, [0, 2] % at 10 stations and [2, 5] % at 3 stations. Our relative RMSEs lie in the interval [1.7, 4.7] % which is a bit less than theirs: [5, 10] % in most cases. We may conclude that both studies offer similar results.





**Table 18. Correlation coefficient, and biases and RMSE relative to the mean of the measurements from Mabasa et al. (2021) for *G*.**

| Station | Correl. coeff. | Rel. bias (%) | Rel. RMSE (%) | Station | Correl. coeff. | Rel. bias (%) | Rel. RMSE (%) |
|---|---|---|---|---|---|---|---|
| Thohoyandou | > 0.995 | [−5, −2] | [5, 10] | Durban | > 0.995 | [2, 5] | [5, 10] |
| Polokwane | > 0.995 | [−10, −5] | [5, 10] | Prieska | > 0.995 | [2, 5] | < 5 |
| Bethlehem | > 0.995 | [2, 5] | [5, 10] | De Aar | > 0.995 | [0, 2] | < 5 |
| Nelspruit | > 0.995 | [2, 5] | [5, 10] | Mthatha | > 0.995 | [2, 5] | [5, 10] |
| Mahikeng | > 0.995 | [2, 5] | [5, 10] | George | > 0.995 | [2, 5] | < 5 |
| Irene | > 0.995 | [2, 5] | [5, 10] | Cape Town | > 0.995 | [−2, 0] | < 5 |
| Upington | > 0.995 | [0, 2] | < 5 | | | | |

### 6.3 Performances reported in previous works

The results reported in other works for *G* and *B* or $B_N$ at summarizations 1 min, 10 min and 1 h are listed in Tables A1 to A6 in Appendix. The stations are different from ours; the objective of this Section is to assess whether our findings are in agreement with similar published works regarding the range of values for each indicator and the variability of these indicators between sites. The comparison is summarized in Tables 19 (*G*) and 20 (*B* or $B_N$). We have excluded the stations Humboldt and Sacramento, and all Egyptian stations but Cairo

and Aswan from our analysis for *G* as they exhibit extreme unexplained values compared to the other stations. Regarding *G*, the correlation coefficients at 1 min (Table A1) and 10 min (Table A2) range between 0.97 and 1.00, except at Xianghe (0.954), and those at summarization 1 h (Table A3) range between 0.94 and 0.99. They are in line with ours: [0.96, 0.99] though a bit greater (Table 19). The bias is positive at most stations regardless of the summarizations and previous results are in line with ours. The standard deviations of errors are

similar between summarizations and this study (Table 19). The range of values is narrow in all cases though narrower in our study in relative value. The intervals of the RMSEs are similar between summarizations and this study (Table 19).

Looking at Table A1, we found that the mean of *G* is correctly estimated at stations located in climates BWh, Cfa, Cfb, and Csa, with a few exceptions in Pakistan due to local aerosol loads in dust or city pollution, and

overestimated in climate Aw (Peshawar, Islamabad, Brasilia), in agreement with our own findings (see Section 4). The agreement is less clear at greater summarizations (Tables A2-A3).

**Table 19. Ranges of correlation coefficients, bias, standard deviations, RMSE and their values relative to the mean of the measurements for *G* for summarizations 1 min, 10 min and 1 h, and those of the present study**

| Station | Correl. coeff. | Bias (W m$^{-2}$) | Rel. bias (%) | Standard deviation (W m$^{-2}$) | Rel. standard deviation (%) | RMSE (W m$^{-2}$) | Rel. RMSE (%) |
|---|---|---|---|---|---|---|---|
| 1 min | >0.97 | [−18, 47] | [−2.3, 6.4] | [17, 37] | [2.1, 6.3] | [18, 58] | [2.3, 7.9] |
| 10 min | >0.99 | [−9, 35] | [−1.4, 6.3] | [26, 31] | [3.2, 5.6] | [22, 47] | [3.6, 5.3] |
| 1 h | [0.94, 0.99] | [−9, 53] | [−1.6, 9.7] | [13, 37] | [2.0, 4.4] | [14, 52] | [2.1, 7.4] |
| **This study** | **[0.96, 0.99]** | **[−20, 45]** | **[−2.3, 5.5]** | **[13, 31]** | **[1.3, 3.7]** | **[16, 63]** | **[1.7, 7.1]** |


The correlation coefficients at 1 min (Table A4) for *B* are greater than those for $B_N$ as expected (see Section 6.1). The correlation coefficients at Los Angeles (Table A4) and Cairo (Table A6) are very low: 0.29 and 0.21. If we exclude the value for Los Angeles, the correlation coefficient at 1 min is in the interval [0.59, 0.94]. The intervals of the correlation coefficient are similar between summarizations and this study (Table 20). Likely

because of the absence of the circumsolar radiation in the direct component in McClear, there are some slight





variations between the previous works themselves and this study, especially regarding the bias. In the previous works at summarizations 1 min and 10 min, the bias is most often negative (Tables A4 and A5), contrary to summarization 1 h (Table A6) and our study where more than 60 % of the stations exhibit positive biases. The ranges of relative standard deviations of errors are only fairly close between summarizations and this study

(Table 20) and our study has the narrowest. The intervals of the relative RMSE are fairly similar and this time, our study has the widest interval.

**Table 20. Ranges of correlation coefficients, bias, standard deviations, RMSE and their values relative to the mean of the measurements for $B$ or $B_N$ for summarizations 1 min, 10 min and 1 h, and those of the present study.**

| Station | Correl. coeff. | Bias (W m$^{-2}$) | Rel. bias (%) | Standard deviation (W m$^{-2}$) | Rel. standard deviation (%) | RMSE (W m$^{-2}$) | Rel. RMSE (%) |
|---|---|---|---|---|---|---|---|
| 1 min ($B$) | >0.96 | [−51, 33] | [−9.7, 5.9] | [25, 80] | [5.2, 15.7] | [32, 82] | [5.2, 16.0] |
| 1 min ($B_N$) | [0.59, 0.94] | - | [−5.2, 5.4] | - | - | - | [4.9, 10.0] |
| 10 min ($B_N$) | [0.76, 0.93] | [−68, 13] | [−8.3, 1.6] | [48, 66] | [5.5, 9.6] | [53, 87] | [6.6, 12.6] |
| 1 h ($B_N$) | [0.21, 0.73] | [−72, 65] | [−8.3, 8.8] | [42, 108] | [4.4, 12.9] | [46, 108] | [5.7, 12.9] |
| **This study ($B_N$)** | **[0.53, 0.90]** | **[−56, 90]** | **[−5.9, 10.7]** | **[31, 70]** | **[3.0, 7.9]** | **[32, 111]** | **[1.7, 13.2]** |


As a whole, one may conclude that our results are in agreement with those of the previous works for $G$ and also for $B_N$ given the limitations of this comparison.

## 7 Conclusion

The main goal of this work was to expand and strengthen knowledge on the quality of the McClear outputs in Sub-Saharan Africa and Western Indian Ocean. To this purpose, one minute measurements from 44 stations were compared to coincident McClear outputs. The stations are located in several climates: tropical climates of monsoon (Am) and savannah (Aw) types, arid and hot or cold climates of steppe or desert types (BSh, BWh, BSk, BWk), temperate climates without dry season and hot or warm summer (Cfa, Cfb), temperate climates with

dry and hot summer (Csa) or dry winter and hot (Cwa) or warm (Cwb) summer. Elevations of half of the stations are greater than 100 m above sea level, up to 1914 m.

It was found that the bias for $G$ is most often positive and ranges between −20 W m$^{-2}$ and 45 W m$^{-2}$, i.e. between −2.3 and 5.5 % in relative value, with a mean of 16 W m$^{-2}$ (1.8 %). Half of the stations exhibit bias in the range [−15, 15] W m$^{-2}$, i.e. [−1.5, 1.5] % in relative value. The variance of $G$ is often underestimated with a ratio of

variances in the interval [0.83, 1.05] with a mean of 0.95. About half of the stations (45 %) exhibit ratios comprised between 0.95 and 1.05. The correlation coefficients between measurements and McClear estimates are greater than 0.960. The slopes of the least-squares fitting lines range between 0.88 and 1.05 and between 0.95 and 1.05 at 25 stations out of 44. The standard deviation of errors exhibits a limited range: [13, 30] W m$^{-2}$, with a mean of 22 W m$^{-2}$ (2.4 %). At each of the 44 stations, the relative standard deviation of errors is less than the

expectations listed in Table 6: if the bias were removed, the McClear outputs would conform to the good quality standard of the World Meteorological Organization. The RMSE lies within [17, 53] W m$^{-2}$, with a mean of 31 W m$^{-2}$ (3.4 %). The greater the mean clearness index at a station, the greater the correlation coefficient and the smaller the standard deviation of errors and the RMSE.





As for $B_N$, the bias is most often positive and ranges between $-48$ W m$^{-2}$ and 62 W m$^{-2}$, i.e. between $-5.4$ and 7.7 % in relative value, with a mean of 11 W m$^{-2}$ (1.4 %). The variance of $B_N$ is often overestimated with a ratio of variances in the interval [0.65, 1.50] with a mean of 1.07. The correlation coefficient between measurements and McClear estimates ranges between 0.532 and 0.896 and is often greater than 0.700. The slopes of the least-squares fitting lines are less than 1 and are often close to 0.80. As a whole, there is an overestimation of the smallest $B_N$ and an underestimation of the greatest ones. The standard deviation of errors is in the range [40, 69] W m$^{-2}$, with a mean of 55 W m$^{-2}$ (6.4 %). At each of the 44 stations, the relative standard deviation of errors is greater than the expectations listed in Table 6: if the bias were removed, the McClear outputs would not conform to the good quality standard of the World Meteorological Organization. The RMSE lies within [42, 88] W m$^{-2}$, with a mean of 63 W m$^{-2}$ (7.4 %). The greater the mean clearness index at a station, the greater the correlation coefficient and the smaller the standard deviation of errors and the RMSE.

These figures for $G$ and $B_N$ are in agreement with previously reported performances of the McClear service and therefore confirm that the performances of McClear are fairly similar worldwide as a whole. However, an in-depth analysis reveals some variability in space and time. Performances in $G$ vary from very little to small from year to year at a given station. As for $B_N$ performances from year to year at a given station vary from small to noticeable. The results show spatial consistency of performances for $G$ and $B_N$ at short to lower mesoscales which was one of the objectives of the inception of version 3 (Gschwind et al., 2019). The variability ranges from noticeable to strong at upper mesoscales and greater, except for the correlation coefficient and standard deviation of errors in $G$ whose spatial variability is small.

This study evidences a link between the bias and ratio of variances in $G$ and climate which is confirmed by our analysis of the results of published works though these findings are limited by the low number of stations and other particular conditions such as elevation. In tropical climates, the mean is underestimated in Am and overestimated in Aw while the variance in $G$ is underestimated. The mean is fairly correctly estimated in arid climates (BSh, BSk, BWh, BWk). The variance is underestimated in steppe (BSh, BSk) and correctly estimated in desert (BWh, BWk). The mean is fairly correctly estimated in temperate climates (Cfa, Cfb, Csa) and noticeably overestimated in climates with a dry winter Cwa and Cwb. The variance is correctly estimated in these climates, except in Cwb due to the high elevation of the stations. Actually, the variance is underestimated at elevation greater than 1300 m whatever the climate. The influence of climate or elevation on errors in $B_N$ is much less marked than for $G$.

The analysis of the influence of other variables on errors suggests a major influence of the AOD. Previously published comparisons made between AERONET measurements and AOD from CAMS show links between errors in CAMS AOD and climates that may partly explain our results as well as those from similar works. Actually, the links between the climates and the errors on $G$ and $B_N$ are indirect and result from combinations of gross errors in aerosol properties modelled in CAMS, gross errors in the exploitation of these properties in the McClear model (as discussed by Gschwind et al., 2019) and other shortcomings in the McClear model regarding reflective properties of the ground and vertical profiles of temperature, pressure, density, and volume mixing ratio for gases as a function of altitude.

This work has established an overview of the performances of the McClear service from this study and other similar published works.




It is suggested to developers of the McClear model to include in outputs of the service estimates of the SSI within the spectral range of the pyranometers and its circumsolar part in order to make further comparisons between McClear outputs and measurements more accurate by *i)* removing the spectral effects, and *ii)* easing the comparison of the direct component.

**8 Appendix: Tables of the results reported in other works for *G* and *B* or $B_N$ at summarizations 1 min, 10 min and 1 h**


The following tables (A1 to A6) list the results reported in other works for *G* and *B* or $B_N$ at summarizations 1 min, 10 min and 1 h.

**Table A1. Results from previous works at summarization 1 min. Means of the measurements, correlation coefficients, bias, standard deviations, RMSE and their values relative to the mean of the measurements for *G*.**

| Station | Letter code | Mean of measurements (W m⁻²) | Correl. coeff. | Bias (W m⁻²) | Standard deviation (W m⁻²) | Rel. bias (%) | Rel. standard deviation (%) | RMSE (W m⁻²) | Rel. RMSE (%) |
|---|---|---|---|---|---|---|---|---|---|
| Barrow (Alaska) | (a) | 498 | 0.990 | −6 | 20 | −1.2 | 4.0 | 21 | 4.2 |
| Toravere (Estonia) | (c) | 484 | 0.995 | 2 | 18 | 0.4 | 3.7 | 19 | 3.9 |
| Cabauw | (c) | 446 | 0.995 | 8 | 21 | 1.8 | 4.7 | 23 | 5.2 |
| (The Netherlands) | (f) | - | 0.995 | - | - | 1.3 | - | - | 3 |
| Palaiseau (France) | (a) | 598 | 0.995 | 7 | 24 | 1.2 | 4.0 | 25 | 4.2 |
| Payerne (Switzerland) | (a) | 596 | 0.995 | 22 | 22 | 3.7 | 3.7 | 29 | 4.9 |
| Carpentras | (a) | 629 | 0.995 | 20 | 21 | 3.2 | 3.3 | 31 | 4.9 |
| (France) | (c) | 553 | 0.995 | 13 | 17 | 2.4 | 3.1 | 22 | 4.0 |
|  | (f) | - | 0.998 | - | - | 2.0 | - | - | 3 |
| Humboldt (California, USA) | (d) | – | – | – | – | 7.5 | – | – | 7.9 |
| Xianghe (China) | (a) | 791 | 0.954 | −7 | 35 | −0.9 | 4.4 | 36 | 4.6 |
| Palma de Mallorca (Spain) | (c) | 567 | 0.990 | 9 | 33 | 1.6 | 5.8 | 34 | 6.0 |
| Burjassot (Spain) | (c) | 596 | 0.990 | 8 | 37 | 1.3 | 6.3 | 38 | 6.4 |
| Sacramento (California, USA) | (d) | – | – | – | – | 6.1 | – | – | 6.6 |
| Hanford (California, USA) | (d) | – | – | – | – | 1.9 | – | – | 4.1 |
| Las Vegas (Nevada, USA) | (d) | – | – | – | – | −0.4 | – | – | 2.4 |
| Tateno (Japan) | (a) | 590 | 0.990 | 10 | 27 | 1.7 | 4.6 | 29 | 4.9 |
| Los Angeles (California, USA) | (d) | – | – | – | – | 1.6 | – | – | 2.9 |
| Peshawar (Pakistan) | (g) | 739 | 0.975 | 47 | 33 | 6.4 | 4.5 | 58 | 7.8 |
| Islamabad (Pakistan) | (g) | 697 | 0.977 | 20 | 34 | 2.9 | 4.9 | 40 | 5.7 |
| Benguerir (Morocco) | (e) | 766 | 0.996 | 5 | 17 | 0.6 | 2.1 | 18 | 2.3 |
| Lahore | (g) | 748 | 0.972 | 19 | 34 | 2.5 | 4.6 | 39 | 5.3 |



| | | | | | | | | |
|---|---|---|---|---|---|---|---|---|
| (Pakistan) | | | | | | | | |
| Sede Boqer | (a) | 785 | 0.995 | 12 | 24 | 1.5 | 3.1 | 27 | 3.4 |
| (Israel) | (c) | 636 | 0.995 | 9 | 28 | 1.4 | 4.4 | 30 | 4.7 |
| Quetta | (g) | 751 | 0.992 | 26 | 23 | 3.5 | 3.0 | 35 | 4.6 |
| (Pakistan) | | | | | | | | |
| Multan | (g) | 741 | 0.973 | 16 | 30 | 2.2 | 3.9 | 34 | 4.5 |
| (Pakistan) | | | | | | | | |
| Bahawalpur | (g) | 720 | 0.981 | −11 | 27 | −1.5 | 3.6 | 29 | 3.9 |
| (Pakistan) | | | | | | | | |
| Khuzdar | (g) | 743 | 0.992 | 17 | 21 | 2.3 | 2.8 | 27 | 3.6 |
| (Pakistan) | | | | | | | | |
| Hyderabad | (g) | 791 | 0.983 | −18 | 26 | −2.3 | 3.3 | 32 | 4.0 |
| (Pakistan) | | | | | | | | |
| Karachi | (g) | 790 | 0.982 | 17 | 25 | 2.1 | 3.2 | 30 | 3.9 |
| (Pakistan) | | | | | | | | |
| Tamanrasset | (a) | 791 | 0.99 | 8 | 18 | 1.0 | 2.3 | 20 | 2.5 |
| (Algeria) | (c) | 650 | 0.995 | 5 | 28 | 0.8 | 4.3 | 28 | 4.3 |
| Sainte-Marie (La | (b) | 821 | 0.99 | – | – | 0.8 | – | – | 3 |
| Réunion Island) | | | | | | | | |
| Brasilia (Brazil) | (a) | 649 | 0.995 | 25 | 24 | 3.9 | 3.7 | 35 | 5.4 |
| Alice Springs | (a) | 715 | 0.995 | 11 | 20 | 1.5 | 2.8 | 23 | 3.2 |
| (Australia) | | | | | | | | |
| Lauder (New | (a) | 600 | 0.995 | 6 | 20 | 1.0 | 3.3 | 21 | 3.5 |
| Zealand) | | | | | | | | |

**Table A2. Results from previous works at summarization 10 min. Means of the measurements, correlation coefficients, bias, standard deviations, RMSE and their values relative to the mean of the measurements for *G*.**

| Station | Letter code | Mean of measurements (W m⁻²) | Correl. coeff. | Bias (W m⁻²) | Standard deviation (W m⁻²) | Rel. bias (%) | Rel. standard deviation (%) | RMSE (W m⁻²) | Rel. RMSE (%) |
|---|---|---|---|---|---|---|---|---|---|
| Beer Sheva (Israel) | (o) | 810 | 0.988 | 19 | 26 | 2.3 | 3.2 | 32 | 4.0 |
| Sede Boqer (Israel) | (o) | 838 | 0.988 | 2 | 30 | 0.2 | 3.6 | 30 | 3.6 |
| Yotvata (Israel) | (o) | 825 | 0.990 | 32 | 26 | 3.9 | 3.2 | 41 | 5.0 |
| Al Aradh (UAE) | (p) | 638 | 0.993 | −9 | 28 | −1.4 | 4.4 | 29 | 4.5 |
| East of Jebel Hafeet (UAE) | (p) | 609 | 0.994 | 9 | 23 | 1.5 | 3.8 | 25 | 4.1 |
| Masdar City (UAE) | (p) | 558 | 0.990 | 35 | 31 | 6.3 | 5.6 | 47 | 8.4 |
| Madinat Zayed #1 (UAE) | (p) | 582 | 0.995 | 18 | 23 | 3.1 | 4.0 | 29 | 5.0 |
| Madinat Zayed #2 (UAE) | (p) | 603 | 0.995 | −5 | 21 | −0.8 | 3.5 | 22 | 3.6 |
| Al Sweihan (UAE) | (p) | 585 | 0.994 | 18 | 25 | 3.1 | 4.3 | 31 | 5.3 |
| Al Wagan (UAE) | (p) | 618 | 0.994 | −1 | 25 | −0.2 | 4.0 | 25 | 4.0 |




**Table A3. Results from previous works at summarization 1 h. Means of the measurements, correlation coefficients, bias, standard deviations, RMSE and their values relative to the mean of the measurements for *G*.**

| Station | Letter code | Mean of measurements (W m$^{-2}$) | Correl. coeff. | Bias (W m$^{-2}$) | Standard deviation (W m$^{-2}$) | Rel. bias (%) | Rel. standard deviation (%) | RMSE (W m$^{-2}$) | Rel. RMSE (%) |
|---|---|---|---|---|---|---|---|---|---|
| Lerwick (United Kingdom) | (v) | 560 | - | −9 | 13 | −1.6 | 2.3 | 16 | 2.9 |
| Toravere (Estonia) | (v) | 500 | - | 12 | 14 | 2.4 | 2.8 | 18 | 3.6 |
| Zilani (Estonia) | (v) | 598 | - | 16 | 25 | 2.7 | 4.2 | 30 | 5.0 |
| Lindenberg (Germany) | (v) | 509 | - | 4 | 17 | 0.8 | 3.3 | 17 | 3.3 |
| Cabauw (The Netherlands) | (v) | 543 | - | 13 | 17 | 2.4 | 3.1 | 21 | 3.9 |
| Valentia (Ireland) | (v) | 618 | - | 12 | 23 | 1.9 | 3.7 | 26 | 4.2 |
| Kassel (Germany) | (v) | 585 | - | 12 | 19 | 2.1 | 3.2 | 22 | 3.8 |
| Wien (Austria) | (v) | 603 | - | 31 | 25 | 5.1 | 4.1 | 40 | 6.6 |
| Bratislava (Slovakia) | (v) | 548 | - | 53 | 23 | 9.7 | 4.2 | 58 | 10.6 |
| Nantes (France) | (v) | 581 | - | 10 | 19 | 1.7 | 3.3 | 21 | 3.6 |
| Kishinev (Moldavia) | (v) | 578 | - | 18 | 16 | 3.1 | 2.8 | 24 | 4.2 |
| Payerne (Switzerland) | (v) | 604 | - | 22 | 17 | 3.6 | 2.8 | 28 | 4.6 |
| Davos (Switzerland) | (v) | 657 | - | −6 | 13 | −0.9 | 2.0 | 14 | 2.1 |
| Geneva (Switzerland) | (v) | 622 | - | 42 | 18 | 6.8 | 2.9 | 46 | 7.4 |
| Vaulx-en-Velin (France) | (v) | 651 | - | 34 | 25 | 5.2 | 3.8 | 42 | 6.5 |
| Carpentras (France) | (v) | 587 | - | 14 | 18 | 2.4 | 3.1 | 23 | 3.9 |
| Ajaccio (Corsica Island) | (t) | - | 0.99 | – | – | −1 | – | – | 4 |
| Madrid (Spain) | (v) | 618 | - | 26 | 23 | 4.2 | 3.7 | 35 | 5.7 |
| Patras (Greece) | (w) | - | 0.99 | 3 | – | 0.5 | – | 30 | 4.9 |
| Almeria (Spain) | (v) | 637 | - | 11 | 15 | 1.7 | 2.4 | 19 | 3.0 |
| Barrani (Egypt) | (u) | 644 | 0.956 | 86 | 43 | 13.4 | 6.7 | 96 | 14.9 |
| Matruh (Egypt) | (u) | 770 | 0.982 | 72 | 30 | 9.4 | 3.9 | 78 | 10.1 |
| Arish (Egypt) | (u) | 741 | 0.941 | 91 | 52 | 12.3 | 7.0 | 105 | 14.2 |
| Sede Boqer (Israel) | (v) | 744 | - | 11 | 24 | 1.5 | 3.2 | 26 | 3.5 |
| Cairo (Egypt) | (v) | 834 | 0.958 | 37 | 37 | 4.4 | 4.4 | 52 | 6.2 |
| Asyut (Egypt) | (v) | 801 | 0.947 | 49 | 44 | 6.1 | 5.5 | 66 | 8.2 |
| Kharga (Egypt) | (u) | 753 | 0.974 | 86 | 33 | 11.4 | 4.4 | 92 | 12.2 |
| Aswan (Egypt) | (u) | 856 | 0.960 | 33 | 35 | 3.9 | 4.1 | 48 | 5.6 |
| Tamanrasset (Algeria) | (v) | 672 | - | 16 | 17 | 2.4 | 2.5 | 23 | 3.4 |





| Station | Letter code | Mean of measurements (W m⁻²) | Correl. coeff. | Bias (W m⁻²) | Standard deviation (W m⁻²) | Rel. bias (%) | Rel. standard deviation (%) | RMSE (W m⁻²) | Rel. RMSE (%) |
|---|---|---|---|---|---|---|---|---|---|
| Mount Kenya (Kenya) | (v) | 377 | - | 0 | 5 | 0.0 | 1.3 | 5 | 1.3 |
| Skukuza (South Africa) | (v) | 631 | - | 28 | 26 | 4.4 | 4.1 | 38 | 6.0 |


**Table A4. Results from previous works at summarization 1 min. Means of the measurements, correlation coefficients, bias, standard deviations, RMSE and their values relative to the mean of the measurements for $B$ or $B_N$.**

| Station | Letter code | Mean of measurements (W m⁻²) | Correl. coeff. | Bias (W m⁻²) | Standard deviation (W m⁻²) | Rel. bias (%) | Rel. standard deviation (%) | RMSE (W m⁻²) | Rel. RMSE (%) |
|---|---|---|---|---|---|---|---|---|---|
| Barrow | (a) | 406 | 0.964 | −21 | 35 | −5.2 | 8.6 | 41 | 10.1 |
| Toravere | (c) | 404 | 0.990 | −21 | 25 | −5.2 | 6.2 | 32 | 7.9 |
| Cabauw | (c) | 333 | 0.97 | 3 | 38 | 0.9 | 11.4 | 38 | 11.4 |
| | (f) ($B_N$) | - | 0.947 | - | - | 5.4 | - | - | 10 |
| Palaiseau | (a) | 492 | 0.980 | −3 | 37 | −0.6 | 7.5 | 37 | 7.5 |
| Payerne | (a) | 505 | 0.980 | 6 | 39 | 1.2 | 7.7 | 35 | 6.9 |
| Carpentras | (a) | 530 | 0.985 | −1 | 35 | −0.2 | 6.6 | 39 | 7.4 |
| | (c) | 465 | 0.990 | −11 | 32 | −2.4 | 6.9 | 34 | 7.3 |
| | (f) ($B_N$) | - | 0.943 | - | - | 1.3 | - | - | 8 |
| Humboldt | (d) ($B_N$) | – | 0.66 | – | – | 4.1 | – | – | 6.5 |
| Xianghe | (a) | 642 | 0.860 | −22 | 60 | −3.4 | 9.3 | 64 | 10.0 |
| Palma de Mallorca | (c) | 460 | 0.970 | −22 | 58 | −4.8 | 12.6 | 62 | 13.5 |
| Burjassot | (c) | 474 | 0.964 | 1 | 61 | 0.2 | 12.9 | 61 | 12.9 |
| Sacramento | (d) ($B_N$) | – | 0.69 | – | – | -0.2 | – | – | 4.9 |
| Hanford | (d) ($B_N$) | – | 0.59 | – | – | 4.1 | – | – | 6.6 |
| Las Vegas | (d) ($B_N$) | – | 0.59 | – | – | -5.2 | – | – | 8.3 |
| Tateno | (a) | 485 | 0.970 | −16 | 41 | −3.3 | 8.5 | 44 | 9.1 |
| Los Angeles | (d) ($B_N$) | – | 0.29 | – | – | −1.6 | – | – | 7.3 |
| Benguerir | (e) ($B_N$) | 766 | 0.845 | −17 | 56 | −1.9 | 6.4 | 58 | 6.7 |
| Sede Boqer | (a) | 667 | 0.975 | −48 | 39 | −7.2 | 5.8 | 62 | 9.3 |
| | (c) | 527 | 0.985 | −51 | 51 | −9.7 | 9.7 | 72 | 13.7 |
| Tamanrasset | (a) | 653 | 0.975 | 16 | 45 | 2.5 | 6.9 | 48 | 7.4 |
| | (c) | 511 | 0.949 | 16 | 80 | 3.1 | 15.7 | 82 | 16.0 |
| Brasilia | (a) | 560 | 0.990 | 33 | 35 | 5.9 | 6.3 | 48 | 8.6 |
| Alice Springs | (a) | 634 | 0.990 | 4 | 33 | 0.6 | 5.2 | 33 | 5.2 |
| Lauder | (a) | 544 | 0.990 | −32 | 36 | −5.9 | 6.6 | 48 | 8.8 |

**Table A5. Results from previous works at summarization 10 min. Means of the measurements, correlation coefficients, bias, standard deviations, RMSE and their values relative to the mean of the measurements for $B_N$.**

| Station | Letter code | Mean of measurements (W m⁻²) | Correl. coeff. | Bias (W m⁻²) | Standard deviation (W m⁻²) | Rel. bias (%) | Rel. standard deviation (%) | RMSE (W m⁻²) | Rel. RMSE (%) |
|---|---|---|---|---|---|---|---|---|---|
| Beer Sheva | (o) | 841 | 0.759 | −46 | 51 | −5.5 | 6.1 | 69 | 8.2 |
| Sede Boqer | (o) | 878 | 0.781 | −68 | 48 | −7.7 | 5.5 | 83 | 9.5 |
| Yotvata | (o) | 809 | 0.794 | 13 | 51 | 1.6 | 6.3 | 53 | 6.6 |
| Al Aradh | (p) | 690 | 0.920 | −57 | 66 | −8.3 | 9.6 | 87 | 12.6 |
| East of Jebel Hafeet | (p) | 683 | 0.922 | −15 | 60 | −2.2 | 8.8 | 62 | 9.1 |
| Masdar City | (p) | 634 | 0.911 | 6 | 62 | 0.9 | 9.8 | 62 | 9.8 |
| Madinat Zayed #1 | (p) | 670 | 0.929 | −25 | 59 | −3.7 | 8.8 | 64 | 9.6 |





| Madinat Zayed #2 | (p) | 681 | 0.926 | −42 | 61 | −6.2 | 9.0 | 74 | 10.9 |
| Al Sweihan | (p) | 660 | 0.922 | −16 | 61 | −2.4 | 9.2 | 63 | 9.5 |
| Al Wagan | (p) | 668 | 0.928 | −41 | 62 | −6.1 | 9.3 | 74 | 11.1 |

**Table A6. Results from previous works at summarization 1 h. Means of the measurements, correlation coefficients, bias, standard deviations, RMSE and their values relative to the mean of the measurements for $B_N$.**

| Station | Letter code | Mean of measurements (W m$^{-2}$) | Correl. coeff. | Bias (W m$^{-2}$) | Standard deviation (W m$^{-2}$) | Rel. bias (%) | Rel. standard deviation (%) | RMSE (W m$^{-2}$) | Rel. RMSE (%) |
|---|---|---|---|---|---|---|---|---|---|
| Lerwick | (v) | 802 | - | −44 | 55 | −5.5 | 6.9 | 70 | 8.7 |
| Toravere | (v) | 806 | - | −9 | 50 | −1.1 | 6.2 | 51 | 6.3 |
| Zilani | (v) | 834 | - | 10 | 47 | 1.2 | 5.6 | 48 | 5.8 |
| Lindenberg | (v) | 782 | - | −12 | 52 | −1.5 | 6.6 | 53 | 6.8 |
| Cabauw | (v) | 758 | - | 10 | 56 | 1.3 | 7.4 | 57 | 7.5 |
| Valentia | (v) | 855 | - | −56 | 54 | −6.5 | 6.3 | 78 | 9.1 |
| Kassel | (v) | 793 | - | 4 | 58 | 0.5 | 7.3 | 58 | 7.3 |
| Wien | (v) | 767 | - | 36 | 63 | 4.7 | 8.2 | 73 | 9.5 |
| Bratislava | (v) | 740 | - | 65 | 63 | 8.8 | 8.5 | 91 | 12.3 |
| Nantes | (v) | 807 | - | 4 | 56 | 0.5 | 6.9 | 56 | 6.9 |
| Kishinev | (v) | 804 | - | 8 | 45 | 1.0 | 5.6 | 46 | 5.7 |
| Payerne | (v) | 819 | - | 13 | 60 | 1.6 | 7.3 | 61 | 7.4 |
| Davos | (v) | 954 | - | −61 | 42 | −6.4 | 4.4 | 74 | 7.8 |
| Geneva | (v) | 814 | - | 29 | 53 | 3.6 | 6.5 | 60 | 7.4 |
| Vaulx-en-Velin | (v) | 817 | - | 25 | 61 | 3.1 | 7.5 | 66 | 8.1 |
| Carpentras | (v) | 820 | - | −2 | 55 | −0.2 | 6.7 | 55 | 6.7 |
| Madrid | (v) | 858 | - | 11 | 48 | 1.3 | 5.6 | 49 | 5.7 |
| Almeria | (v) | 854 | - | −24 | 47 | −2.8 | 5.5 | 53 | 6.2 |
| Sede Boqer | (v) | 868 | - | −72 | 50 | −8.3 | 5.8 | 88 | 10.1 |
| Cairo | (v) | 766 | 0.205 | 31 | 91 | 4.0 | 11.9 | 96 | 12.5 |
| Aswan | (u) | 830 | 0.727 | −21 | 59 | −2.5 | 7.1 | 63 | 7.6 |
| Tamanrasset | (v) | 850 | - | 20 | 62 | 2.4 | 7.3 | 65 | 7.6 |
| Mount Kenya | (v) | 834 | - | 3 | 108 | 0.4 | 12.9 | 108 | 12.9 |

## 9 Code availability

The various codes used for the comparison and plots implement well-known equations for computation of differences and statistics and well-known libraries in Matlab and Python and offer no specificities. McClear-v3 is available as a model with all elements available on the same ftp site than v2 (ftp://ftp.oie-lab.net/pub/, last accessed: 2021-07-23).

## 10 Data availability

All data used in this research can be freely accessed through several public sources.

The BSRN data are the LR0100 product. They can be accessed freely upon registration at https://bsrn.awi.de, last access: 2022-08-23.

The SAURAN network offers free access to its measurements via the web site https://sauran.ac.za/, last access: 2022-08-12.

The World Bank and the International Finance Corporation, collectively the World Bank Group has developed an open data platform energydata.info that provides access to a large number of datasets of solar radiation at



surface. The other measurements used in this work originate from this resource (https://energydata.info/dataset, last accessed on 2021-08-08).

The McClear outputs and inputs from CAMS can be accessed freely upon registration at the CAMS Radiation Service (http://www.soda-pro.com/web-services/radiation/cams-mcclear, last access: 2022-07-30).

**11 Author contribution**

The concept and design of the work were made by WWN and LW. WWN, YMSD, AA and LW designed the methodology. WWN and YMSD identified sources of measurements from the stations, collected them and performed their curation. WWN performed the plausibility check and the filtering of clear-sky instants. WWN collected the McClear estimates and performed the statistical analysis together with YMSD. LW did the analysis of similar previous works. All co-authors analyzed the results and contributed to the discussion. LW wrote the

first draft with help from WWN and YMSD. WWN wrote the subsequent versions with contributions from all co-authors.

**12 Competing interests**

The authors declare that they have no conflict of interest.

**13 Acknowledgements**

The research leading to these results has partly received funding from the Copernicus Atmosphere Monitoring Service, a program being operated by the European Centre for Medium-Range Weather Forecasts (ECMWF) on behalf of the European Union. The authors thank all operators of the ground stations whose valuable measurements they use in their work and the Alfred-Wegener Institute, the SAURAN network and the World Bank Group for hosting and maintaining the websites from which measurements can be collected.

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
