# Peer review of "Further validation of the estimates of the downwelling solar radiation at ground level in cloud-free conditions provided by the McClear service: The case of the Sub-Saharan Africa and Maldives Archipelago"

_EGUsphere, 2022_

## Referee Comment (RC1)

The study validates McClear model against 1 minutes Global Horizontal Irradiance (G) and Direct normal Irradiance ($B_N$) in areas where validation was never carried out before. The study reviewed literature related and followed the same procedures for easy comparison of the results. The article is well written and covers some research gaps in the previous studies which validated McClear model in different regions.

Minor comments

Line 18: Please put brackets on (G)

Line 19 : please put bracket on ($B_N$)

Please be consent with abbreviations to refer Global Horizontal Irradiance (G) and Direct normal Irradiance ($B_N$) so that the readers will not be confused , use those one through out the article not SSI to refer Global Horizontal Irradiance as used in the abstract.

Line 58: Please replace AOD 1020nm with AOD 1240nm since it is the input to the model.

Line 92: Please paraphrase the sentence , the word **'or'** is and the wrong place , making the statement to be confusing

Line 97 :Please paraphrase the sentence , the are a lot of **'and'**, making in unclear validated the model in which country or region.

Line 120: Please add **horizontal** between direct and component because there is also direct normal component, so that the two components or parameters will be differentiated.

Line 219: Can you please summarise or elucidate how the visual check was implemented and applied , some practitioners or researchers might want to apply it in their studies as well.

On the methodology to differentiate between night and day values its not clear which procedure was used, most studies remove everything that falls in a solar zenith angle greater than 85 degrees, this helps to filter pyranometers and pyrheliometers noise and it influence the overall mean of G and $B_N$. How did you calculate the mean averages of G and $B_N$ like the ones given in Table 5 ?

Figure 3 in page 15: The 2 2D graphs seems like you only used G and $B_N$ values greater than 600 w/m2 ? was there no values less than 600 W/m2 in your comparison ?

---

## Author Comment (AC1)

**ANSWERS TO REFEREE #1**

First of all, we thank Referee #1 for these positive remarks and comments on this topic. The comments have been addressed below and have been taken into account for revising a part of the text following recommendations of the referee. The responses to the referee points are below after the reviewer points that are in italics.

**'Comment on egusphere-2022-1023', Brighton Mabasa**

*The study validates McClear model against 1-minute Global Horizontal Irradiance (G) and Direct normal Irradiance ($B_N$) in areas where validation was never carried out before. The study reviewed literature related and followed the same procedures for easy comparison of the results. The article is well written and covers some research gaps in the previous studies which validated McClear model in different regions.*

We thank you for your positive comments on the manuscript.

*Minor comments*

*Line 18: Please put brackets on (G)*

Done as requested. Thanks.

*Line 19: please put bracket on ($B_N$)*

Done as requested. Thanks.

*Please be consent with abbreviations to refer Global Horizontal Irradiance (G) and Direct normal Irradiance ($B_N$) so that the readers will not be confused, use those one throughout the article not SSI to refer Global Horizontal Irradiance as used in the abstract.*

Thanks for this remark. We have defined the surface solar irradiance, abbreviated as SSI, as the irradiance received on a horizontal surface. We further used this abbreviation SSI as a general term when there was no possible confusion. We used the variable $G$ in more specific cases. According to your remark, we have screened again our text and made a few change to be more precise and avoid further confusion. Note that we have preferred to use the abbreviation SSI instead of GHI because the former is used in many domains while GHI is mostly used in the domain of solar energy.

*Line 58: Please replace AOD 1020nm with AOD 1240nm since it is the input to the model.*

Thanks for this remark. Done as requested.

*Line 92: Please paraphrase the sentence, the word '**or**' is and the wrong place, making the statement to be confusing*

Thanks for this remark. Done as requested.

*Line 97: Please paraphrase the sentence, they are a lot of '**and**', making in unclear validated the model in which country or region.*

Thanks for this remark. We have rewritten the relevant part accordingly as follows:

"Dev et al. (2017) performed a comparison in Singapore while Zhong and Kleissl (2015) performed their own in California"

*Line 120: Please add **horizontal** between direct and component because there is also direct normal component, so that the two components or parameters will be differentiated.*

Thanks for this remark. We have rewritten this part of the text as follows:

"One-minute ground-based measurements of irradiance received on a horizontal surface, namely the global irradiance $G$, its diffuse component $D$ and its direct component $B$, or the direct component received at normal incidence $B_N$,"

*Line 219: Can you please summarise or elucidate how the visual check was implemented and applied, some practitioners or researchers might want to apply it in their studies as well.*

There was no specific tool to perform a visual check. We have brought this precision:

"Then, time series of the retained measurements were plotted together with the corresponding irradiances at the top of the atmosphere and a visual check was performed to detect and scrutinize outliers that are possibly rejected. "

*On the methodology to differentiate between night and day values its not clear which procedure was used, most studies remove everything that falls in a solar zenith angle greater*

*than 85 degrees, this helps to filter pyranometers and pyrheliometers noise and it influence the overall mean of G and BN. How did you calculate the mean averages of G and BN like the ones given in Table 5 ?*

Thanks for this remark. The means reported in Table 5 were computed only for selected clear–sky instants after applying all criteria listed in the subsection 3.1. Their combinations remove systematically night values and filter out values for large solar zenith angle. In order to make clearer that we are dealing only clear-sky instants, we have changed the sentence:

"Only these 1 min clear-sky instants were retained for the validation."

to:

"Only these 1 min clear-sky instants were retained for the validation and all computations in the following were made with this subset of clear-sky instants."

*Figure 3 in page 15: The 2 2D graphs seems like you only used G and BN values greater than 600 w/m2 ? was there no values less than 600 W/m2 in your comparison ?*

Thanks for this remark. Of course, raw measurements include irradiances lower than 600 W m$^{-2}$. After applying all criteria listed in the subsection 3.1, the selected clear–sky instants are those with irradiances greater than 600 W m$^{-2}$.

---

## Author Comment (AC2)

**ANSWERS TO REFEREE #2**

First of all, we thank Referee #2 for these constructive remarks and comments on this topic. The comments have been addressed below and have been taken into account for revising a part of the text following recommendations of the referee. The responses to the referee points are below after the reviewer points that are in italics.

**'Comment on egusphere-2022-1023', Anonymous Referee #2**

*General comments*

*This paper presents a validation exercise of McClear across Sub-Saharian Africa and the Maldives Archipelago. The study includes observations from several stations that have never been used, or just occasionally, for similar studies, and thus provide valuable information for users and developers of McClear.*

We thank you for your positive overview on our work.

*The paper is too long. At times, it is difficult to read because it is profuse in details. For instance, it replicates in the text many data that is already in the tables. Surely, many should have been removed to make the paper more concise and shorter. It also discusses aspects of the study that could have been omitted.*

We have made efforts to shorten the text by rewriting some parts here and there, removing redundant information, suppressing several details by referring to the original works, and reducing comments on graphs and tables. One page or so is saved doing so.

We have also created a new annex where we have placed several graphs (Figs. 4 to 8) which are not of uttermost importance to understand the work at first reading, thus reducing the length of the main text by about five pages.

*The language is correct and the scientific quality is good.*

We thank you for these positive comments.

*As a general comment, I would like to add that McClear is not run typically as an usual model that is decoupled from the inputs that are used to run it. Instead, it is normally used as a web service in combination with inputs from CAMS, despite it could also be run with other input sources. This is an important fact because, indeed, talking about "validation of the McClear estimates" normally hides the subtlety that the validation is of McClear + CAMS. Hence, I have always thought that a better naming convention would be McClear Service to clearly state that this is the modelling approach taken, and not other.*

We fully agree. We clearly stated in abstract and text that we are dealing with the McClear service which is based on the McClear model. The McClear service provides estimates of downwelling solar radiation at ground level in cloud-free conditions and these estimates are compared to ground-based measurements. The comment from Referee #2 shows that some confusion was still possible. To eliminate it, we have rewritten several parts by using the words "McClear service" thus making it clear that it is "McClear+CAMS+other sources". (Note that there are several other sources than CAMS that are automatically accessed to by the McClear service).

We also changed the title of the article from:

"Further validation of the McClear estimates of the downwelling solar radiation at ground level in cloud-free conditions: The case of the Sub-Saharan Africa and Maldives Archipelago"

to:

"Further validation of the estimates of the downwelling solar radiation at ground level in cloud-free conditions provided by the McClear service: The case of the Sub-Saharan Africa and Maldives Archipelago."

*I wonder why you chose to validate global and direct irradiance, but not diffuse. Although, admittedly, global and direct irradiances are likely more important for practical applications,*

*the validation of diffuse irradiance is also important from the point of view of model development. The best models parameterize direct and diffuse irradiances independently and evaluate global from them. Hence, the importance of validating diffuse irradiance.*

As mentioned by Referee #2, the McClear model v3 estimates the direct and diffuse components independently and sums these components to obtain the global irradiance. We agree that the global irradiance and its direct component have practical applications. As measurements comprise the diffuse component, it would have been possible to present results for the diffuse component in addition to those for the global or direct or in replacement of the global. The paper is long and we cannot present the results for the global and its two components: we had to make a choice. We arbitrarily decided to present the results for global and direct because of the practical applications above-mentioned and because previous similar works dealt with the global and direct to which we could not easily compare had we done otherwise.

*One final general comment is related with the cloud screening algorithm used here. Nothing against it, but just against the claim that it provides more confident values than the Long and Ackerman algorithm. Based on my own experience, I see difficult that the simple Lefèvre et al approach can cope with the milliard of different sky situations that may hide clear skies, specially in a region like the one considered here. And I don't know either if the Long and Ackerman approach can do it, at least, without a previous calibration of their empirical coefficients (they were set for a totally different environment).*

We agree with Referee #2 that automatically detecting cloud-free situations is very difficult due to "the milliard of different sky situations". The claim "Lefèvre et al. (2013) have found that the results of their algorithm provide less low values of SSI than that of Long and Ackerman (2000) and therefore wrote that their algorithm offers more confidence in the fact that the instant is clear." has been taken from Lefèvre et al. (2013) and is subject to debate which is out of the scope of this article. It is not our intention to discuss such algorithms and we should have not adopted this statement as ours. We have no definitive opinion on the best algorithm and we recognize that we have taken the Lefèvre et al. algorithm because it is that used by the developers of the McClear service in their validation and therefore it allows a comparison between our work and theirs.

We have rewritten the following paragraph:

"A screening algorithm needs to be applied on the ground measurements to separate the cloud-contaminated instants from the cloud-free ones. Several algorithms for detecting clear-sky instants from measurements have been published (see e.g. Bright et al., 2020; Calbó et al., 2001; Ellis et al., 2019; Long and Ackerman, 2000; Reno and Hansen, 2016). Here, the algorithm of Lefèvre et al. (2013) was selected. Lefèvre et al. (2013) have found that the results of their algorithm provide less low values of SSI than that of Long and Ackerman (2000) and therefore wrote that their algorithm offers more confidence in the fact that the instant is clear. The possible influence of the algorithm for detecting clear-sky instants on results is discussed in Section 6."

to:

"Several algorithms for detecting clear-sky instants from measurements have been published (see e.g. Bright et al., 2020; Calbó et al., 2001; Ellis et al., 2019; Long and Ackerman, 2000; Reno and Hansen, 2016). Here, the algorithm of Lefèvre et al. (2013) was chosen because it is that used by the developers of the McClear service in their validation. The possible influence of the algorithm on results is discussed in Section 6."

***Specific comments***

*P1L20. The correlation coefficient is not very much significant or relevant in clear-sky models because the cloudless irradiance is highly determined by a deterministic signal.*

Yes, we agree with Referee. Nevertheless it is a quantity that is expected by many practitioners in a validation exercise and this is why we have presented it. At least, it may indicate that the variations in aerosols and other variables influencing the irradiance in cloudless conditions are either well reproduced by CAMS and other sources or are negligible with respect to the irradiance.

*P1L21. What do you mean by "correctly estimated"? What is "correctly"?*

Thanks for this remark. It is our mistake. We would like to say "accurately". and we modified accordingly.

*P1L23. "relative bias" respect to what?*

Thanks for this remark. We have added the precision as follows: "relative to the means of the measurements at each station".

We have also brought this precision in Section 3.3 "Methodology of validation"

*P2L38. Why that precise range for the shortwave spectrum? Can you provide justification?*

Thanks for this remark. We found that it is not relevant to mention it here since the solar spectrum is larger than that. Therefore, it has been removed.

*P2L39-40. "Other terms… incoming shortwave radiation". Most of these terms are _not_equivalent to SSI. SSI is an irradiance, that is, density flux of energy. It is not clear to me that solar exposure, or solar insolation does precisely refer to the same concept. Please, clarify.*

Thanks for this remark. It is our mistake as "Solar exposure, solar insolation" are terms used in the World Meteorological Organization for irradiation. We have kept appropriate terms and removed others as follows: "Other terms may be found in the literature, such as solar flux, downwelling solar irradiance at the surface, downwelling shortwave flux, or surface incoming shortwave irradiance."

*P2L41. "appearing to come"? There may be scattered photons that "appear to come" from the direction of the sun Would you say that such photons are contributing to direct irradiance or they contribute to diffuse irradiance? Clearly, they are part of diffuse irradiance simply because they have been scattered. What defines direct and diffuse are the extinction processes in the atmosphere.*

Thank you for this remark. The sentence was unclear. There are several definitions of the direct component (see e.g. Blanc, P., Espinar, B., Geuder, N., Gueymard, C., Meyer, R., Pitz-Paal, R., Reinhardt, B., Renne, D., Sengupta, M., Wald, L., and Wilbert, S.: Direct normal irradiance related definitions and applications: the circumsolar issue, Sol. Energy, 110, 561-577, https://doi.org/10.1016/j.solener.2014.10.001, 2014.)

The original sentence "Briefly speaking, the radiation appearing to come from the direction of the sun is the direct component, noted *B*, and the diffuse component gathers the photons coming from the other directions of the sky, noted *D*." has been rewritten as:

"Roughly speaking, the radiation measured on a horizontal surface looking in the direction of the sun is the direct component, noted *B*, while the diffuse component, noted *D*, is the sum of the fluxes coming from the other directions of the sky and impinging on this surface."

*P2L47. "It depends on… these variables define the solar radiation impinging on a horizontal surface…" I think this is a tautology.*

We have replaced the sentence:

"It depends on the date and time of the day and geographic coordinates because these variables define the solar radiation impinging on a horizontal surface at the top of the atmosphere and the solar zenithal and azimuthal angles."

by

"It depends on the date and time of the day and geographic coordinates.".

*P7L179. Can you describe exactly the meaning of "95 % probability"?*

Thanks for this remark. We would like to say "95% confidence level". The part of the text has been rewritten accordingly.

*P8L243-247. "Sea salt and dust… in and below the clouds" Is it necessary to add these comments here? They refer to the treatment of aerosols in CAMS, and it may be obscure for some readers without better context and clarification.*

Thanks for this remark. We have removed this detailed information. More generally, this section was suppressed in order to decrease the length of the text and the reader is referred to Gschwind et al. (2019) for such details.

*P9L248. "resampled in time" How? I presume it is not the same when one goes from 3 hours to 1-min time steps, than when going from 3 hours to daily time steps, for instance.*

Thanks for this remark. We agreed with you. This section was suppressed in order to decrease the length of the text and the reader is referred to Gschwind et al. (2019) for such details. However, we have brought this precision at the end of this section and wrote:

"In the verbose mode, the flow returned by the service contains 1 min values of readings from CAMS resampled to the selected location by spatial bilinear interpolation and resampled in time by linear interpolation, namely, the optical depth of aerosols at 500 nm, and the total column contents in water vapor and ozone."

*P9L255-257. "If not provided, ... cell is taken into account" Specifically, how is the elevation difference accounted for?*

Thanks for this remark. This section was suppressed in order to decrease the length of the text and the reader is referred to Gschwind et al. (2019) for such details. However, how the elevation difference is accounted for was described in Section 5.2 "Uncertainties of the inputs to McClear and of the McClear model itself" (lines 705-707 in the original text) and is kept in the revised version..

*P9L257-258. "The yearly average... total solar irradiance noted E_TSI" This sentence appears to confuse the concepts of total solar irradiance and solar constant. I suggest to review these papers: https://doi.org/10.1016/j.solener.2018.04.001, and https://doi.org/10.1016/j.solener.2018.04.067*

Thank you for the two references. We agree that the sentence is confusing and that it was not clear that we were referring to the solar constant. This section was suppressed in order to decrease the length of the text and the reader is referred to Gschwind et al. (2019) for such details.

Though not mentioned by Referee #2, we replaced "solar total irradiance" by "solar constant" in Section 5.2 "Uncertainties of the inputs to McClear and of the McClear model itself".

*P10L286-287. "...the results of their algorithm provide less low values of SSI... offers more confidence..." Do you think this is a true argument to assign more confidence? Then, it is easy to create a cloud screening algorithm that offers more confidence than that of Lefevre et al: simply retain even less low values of SSI.*

This remark joins that made in the general comment on the cloud screening algorithm. We believe that our revised version answers that comment.

*P10L291. Wrong reference Ineichen and Perez (1999). The correct one is Perez et al, 1990: Making full use of the clearness index for parameterizing hourly insolation conditions. Solar Energy, Vol. 45, No. 2, pp. 111-114.*

Thanks for this remark. We have replaced it accordingly throughout the manuscript.

*P10L300. "… the Rayleigh atmosphere" Strictly speaking, the scale height is rather obtained assuming a hydrostatic atmosphere at 288 K. That is, I might have a Rayleigh atmosphere (i.e., only molecules) that is not hydrostatic. Then, that height scale figure would not be theoretically correct.*

Thanks for this remark. As there is no specific need in this article to discuss the value of this constant 8435.2 used in a formula given by previous authors, we have shortened the sentence from : "… where $z$ is the elevation above sea level expressed in m, and 8435.2 m is the scale height of the Rayleigh atmosphere." to "… where $z$ is the elevation above sea level expressed in m."..

*P10L302. "atmospheric transmission when the reflection of the ground is null" Not clear. As defined in Eq (9), KT includes reflections from the ground.*

Thanks for this remark. In other words, we would like to say, "when there is no reflection of the ground". We have rewritten the text accordingly.

*P14L385. What is the added value of validating KT and KT_BN provided that G and B_N are being also validated?*

Thanks for this remark. G and B_N may offer great variations in value depending on the solar zenithal angle while KT and KT_BN are less dependent. They are also less dependent on the day of the year. Hence, they may bring additional information though we admit that it is small in the case of clear-sky conditions. Actually, we consider validating KT and KT_BN as a means to check our comments and conclusions in the discussion. Note that KT_BN is equal to

KT_B and that KT_BN may bring insights on the performances regarding the direct component on a horizontal surface.

*P14L397. "Knowing that the relative standard deviation is half the relative uncertainty" Why? Can you elaborate more on this?*

Thanks for this remark. The relative uncertainty is the 95 % confidence level, which is equal to twice the standard deviation in the Gaussian case. We have brought the following clarification: "Knowing that the relative standard deviation is half the relative uncertainty assuming a Gaussian distribution of errors, …"

*P28L635. "[240, 4606] nm" Why that upper limit? The lower bound is arguably set by ozone absorption. However, is there a strict limit for the shortwave spectral range?*

Thanks for this remark. Since that the development of McClear involve the Kato et al. (1999) approach as described in section 2.3, the upper limit given by the approach is well 4606 nm. To make it much clearer, we have rewritten this part of the text as follows:

"McClear provides total irradiance, more exactly the irradiance integrated over the [240, 4606] nm range used in the Kato et al. (1999) approach, while measurements by pyranometers are taken in a more limited range, often called broadband range, which is around [285, 2800] nm for pyranometers used in the BSRN, SAURAN and other networks."

*P28L654-658. I don't think you can get a definite conclusion out of it. It can all be just a coincidence that results from the combination of multiple sources of errors that you do not have under control.*

We understand the comment. We have softened our stance and the sentence now read:

"From this, it appears that part of the discrepancies between ground-based measurements and McClear outputs may be attributed to the narrower spectral range of the ground-based instruments, though it cannot be excluded that our observation may be a coincidence resulting from the combination of multiple sources of errors not under control as underlined by an anonymous referee"

*P28L669. "The variance should also be underestimated". I try to understand this sentence, but I am not sure if I did. Apparently you mean that adding DNI, which has some variability, and circumsolar, which also has variability, should result in a signal with higher variability than the two components separately. Hence, if circumsolar is neglected, you will underestimate the variability. However, it does not have to be like this necessarily. What happens is that you are neglecting the correlation between DNI and circumsolar and, indeed, they probably are very much anti-correlated because, for instance, an increase of AOD will reduce DNI, but most often will increase circumsolar.*

Thanks for this remark. We partly disagree with this analysis. Indeed, there a degree of anti-correlation between $B_N$ and the circumsolar radiation and their covariance is not null. However from the best of our knowledge, no work has demonstrated that their correlation coefficient is equal to $-1$, meaning that the variance of the sum is greater than the variance of $B_N$. To take into account the comment of Referee #2, we have changed the sentence into:

 "The variance should also be underestimated though there are some anti-correlation between $B_N$ and the circumsolar radiation."

*Figure 9. The y-axis limits must be adjusted to the range of values that are plotted. As it is now, the figure is totally useless.*

Thanks for this remark. We have updated the figure accordingly

***Technical corrections***

*P1L18. "The global irradiance, G, and…" instead of "The global irradiances G and…"*

Thanks for this remark. Done as requested.

*P1L25. "...the mean of B_N." instead of "...the means of B_N"*

Thanks for this remark. Done as requested.

*P8L231. What is "katoandwandji"? Is it a typo?*

Thanks for this remark. No, it is not a typo. This is given name in libRadtran. We have rewritten this part of the text as follows:

"*katoandwandji* as named in libRadtran"

*P9L270 Is it "v1/v2" a typo?*

Thanks for this remark. To make it much clearer, we have rewritten as follows:

"McClear-v2 and McClear-v3"

*P10L287. Awkward use of "wrote"*

Thanks for this remark. Actually, this paragraph has been modified as discussed in the general comment on cloud screening.

*P32L779. "narrow" instead of "narrow to narrow"?*

Thanks for this remark. Done as requested.